# TOPOLOGY-AWARE GRAPH DIFFUSION MODEL WITH PERSISTENT HOMOLOGY

## ABSTRACT

Generating realistic graphs presents challenges in estimating accurate distribution of graphs in an embedding space while preserving structural characteristics such as topology. However, existing graph generation methods primarily focus on approximating the joint distribution of graph nodes and edges, overlooking topology-wise similarity hindering accurate representation of global graph structures such as connected components and loops. To address this issue, we propose a topology-aware diffusion-based graph generation method that aims to closely resemble the structural characteristics of the original graph by leveraging persistent homology from topological data analysis (TDA). Specifically, we suggest a novel loss function, Persistence Diagram Matching (PDM) loss, which ensures the generated graphs to closely match the topology of the original graphs, enhancing their fidelity and preserving essential homological properties. Also, we introduce a novel topology-aware attention to enhance the self-attention module in the denoising network. Through comprehensive experiments, we demonstrate the effectiveness of our approach not only by exhibiting high generation performance across various metrics, but also by demonstrating a closer alignment with the distribution of topological features observed in the original graphs. In addition, application to real brain network data showcases its versatility and potential for complex and real graph application.

## 1 INTRODUCTION

The major goal of graph generation is to achieve high resemblance between generated graphs and their reference counterparts. To achieve this goal, various graph generation approaches have been taken based on conventional generative models, e.g., recurrent neural networks (You et al., 2018b), variational autoencoders (Simonovsky & Komodakis, 2018) and diffusion models (Jo et al., 2022; Vignac et al., 2023), and each exhibited promising results. Despite the achievement in the context of the quantitative measures, e.g., similarity in distributions of graph characteristics such as degree and clustering coefficients, there remains a limitation on generating graphs coherent to the graph structure via the lens of graph topology.

A brain network is perhaps a suitable example that demonstrates the challenges above. A brain network is typically represented as a graph that characterizes intricate wiring system of the brain, which is comprised of anatomical regions of interest (ROIs) defining its nodes and the connectivity between different ROIs serving as edges (Farahani et al., 2019; Bullmore & Sporns, 2009). It is often large and dense, but its topological properties are well-known as critical biomarkers (Sizemore et al., 2019; Saggar et al., 2018). Moreover, brain networks are expensive; acquiring diffusion magnetic resonance images (dMRI) and processing them via tractography (Sporns et al., 2005) to obtain structural brain connectome data is costly in both cost and labor. In this regime, generating realistic graphs (e.g., brain networks) that preserve their inherent connectivity as well as global structures is highly demanding, however, existing methods often fall short in capturing the essential topological features crucial for modeling interconnected brain regions with high fidelity.

In recent years, diffusion methods have been heavily studied for graph generation (Liu et al., 2023). The methods in Niu et al. (2020) and Jo et al. (2022) proposed score-based diffusion methods in a continuous time domain, originally defined for images (Song et al., 2020). However, the continuous diffusion methods suffer from high computational cost, as the forward and reverse diffusion process is performed on infinitesimal continuous time point. Moreover, the uniformly added Gaussian noise

results in a noisy and complete graph, which causes the loss of structural information and destroys the sparsity of a graph. Later, Vignac et al. (2023) proposed a discrete diffusion method and applied additive noise to each nodes and edges independently for graphs, nevertheless, existing methods overlook the *topologically invariant characteristics*, e.g., geometric shape and connectivity, limiting the generation.

To overcome such issues, we propose a novel **T**opology-**A**ware **G**raph **G**eneration (TAGG) method, from which the sampled graphs resemble not only in the distributions of the original graphs in the embedding space but also in the *homological features* of the original graphs. Conventionally, topological data analysis (TDA) from algebraic topology have been studied in various graph analyses (Carlsson, 2020; Bukkuri et al., 2021; Xu et al., 2021) to investigate topological features, and we bridge the gap between TDA and graph diffusion model to generate topologically realistic graphs. We define Persistence Diagram Matching (PDM) loss with persistence homology, which regularize the homological features of the reference graphs to be incorporated in graph generation process via 1-Wasserstein distance. Furthermore, we introduce a topology-aware attention module, utilizing the homological features derived from the persistence landscape (Bubenik, 2015) of a given graph, to foster the denoising network with global structural information.

**Contributions.** To this end, our main contributions are as follows: **1)** We propose a novel topology-aware graph generation method that yields homologically similar graphs with high fidelity. **2)** We propose PDM loss, utilizing persistent homology to encode the graph topology. **3)** We propose a topology-aware attention module that leverages persistence landscape to enhance the denoising network in capturing graph topology.

Our model demonstrates superior performance on real and synthetic graph generation tasks, with intuitive visualizations for topological comparisons. Especially with the application on brain network generation from Alzheimer's Disease Neuroimaging Initiative (ADNI), our method demonstrates its adaptability to diverse real-world graph generation tasks.

## 2 RELATED WORK

**Graph Generation.** Graph generation has been developed in two major branches; autoregressive and one-shot. Auto-regressive methods (You et al., 2018a;b; Simonovsky & Komodakis, 2018; Jin et al., 2020; Kong et al., 2023; Bergmeister et al., 2024) recursively capture the intricate graph dependencies, and sequentially generate the graph structure conditioned on the current incomplete graph. In spite of their impressive performance, auto-regressive approaches exhibit considerable computational demands due to the increasing number of generation steps along with the graph size. Also, they face a challenge stemming from the absence of an inherent node generation order.

Conversely, one-shot methods (Ma et al., 2018; De Cao & Kipf, 2018; Madhawa et al., 2019; Zang & Wang, 2020) generate the whole graph, i.e., every nodes and edges, at once. By doing so, they reduce the computational requirements while facing performance degradation as the dataset scale grows. Recently, diffusion-based methods (Jo et al., 2022; Vignac et al., 2023; Bergmeister et al., 2024) showed promising capability in graph generation, by defining the forward and reverse diffusion processes and training a neural network that mimic the reverse process to reconstruct the graphs.

**Persistent Homology.** Persistent homology from computational topology studies the topological features of given objects, such as the number of holes (Edelsbrunner & Harer, 2022). It provides a way to capture and quantify the shapes and global structures by computing homological features of objects. By treating a graph as a topological object, the concept of persistent homology can be utilized to analyze the global structure of graphs, which leads improvements for graph classification (Hofer et al., 2020; Zhang et al., 2022; Horn et al., 2022) and link prediction (Yan et al., 2021). Furthermore, persistent homology has been successfully applied in biology (Chen & Volić, 2021; Qiu & Wei, 2023; Bukkuri et al., 2021), signal processing (Xu et al., 2021), and point cloud (Nishikawa et al., 2024), demonstrating its versatility and robustness.

## 3 PRELIMINARIES

We provide a brief introduction to persistent homology, which extracts homological properties from objects. We refer the readers to Edelsbrunner & Harer (2022) and Carlsson (2009), should further questions regarding Topological Data Analysis.

**Simplicial Complex.** Let $V$ be a non-empty set. A simplicial complex $K$ is a collection of non-empty subsets of $V$ which satisfies the following two properties; (1) for any $v \in V$, $\{v\} \in K$, and (2) if $\sigma \in K$ and $\tau \subseteq \sigma$, then $\tau \in K$. An element of $K$ is called a *simplex* and the dimension of a simplex is determined by the length of its elements. For example, an element $\tau \in K$ with $|\tau| = k + 1$ is a $k$-simplex whose dimension is $k$. The dimension of a simplicial complex $K$ is defined by the highest-dimension of its simplices.

**Graph as a Simplicial Complex.** Consider an undirected graph $G = (V, E)$, where $V$ is a set of $N$ nodes and $E \subseteq V \times V$ is a set of edges. Then, a graph $G$ can be interpreted as a 1-dimensional simplicial complex whose 0-simplices are the nodes and 1-simplices are the edges, i.e.,

$$G = K_G = \{\{v\} : v \in V\} \cup E. \tag{1}$$

**Homology.** To investigate the homological properties of a given set $X$ algebraically, assign a chain structure $C_0, C_1, ...$ to $X$ connected by homomorphism $\partial_{k+1} : C_{k+1} \to C_k$, satisfying $\partial_k \circ \partial_{k+1} \equiv 0$ for every integer $k$ (Edelsbrunner & Harer, 2022). We can obtain homological properties of $X$ in $k$ dimension by investigating the homology group $H_k = \ker \partial_k / \mathrm{im} \partial_{k+1}$, where $\ker$ and $\mathrm{im}$ denote the kernel and image of the homomorphism, respectively. The rank of the homology group $H_k$, i.e., Betti number $\beta_k$, represents the homological properties of $X$ in $k$-dimension, e.g., $\beta_0 = \mathrm{rank}(H_0)$ represents the number of connected components, and $\beta_1 = \mathrm{rank}(H_1)$ represents the number of loops.

**Filtration.** Filtration of a graph $G$ is a sequence of nested subgraphs of $G$, i.e., $\emptyset = G^{(0)} \subseteq G^{(1)} \subseteq G^{(2)} \subseteq ... \subseteq G^{(N-1)} \subseteq G^{(N)} = G$. Specifically, the filtration of $G$ can also be defined using a 0-simplex (i.e. vertex) filter function $f : V \to [0, \infty)$ defined as Hofer et al. (2017):

$$\forall \{v'\} \in G, \quad f(\{v'\}) := \deg(\{v'\}) / \max_{\{v\} \in G} (\deg(\{v\})). \tag{2}$$

To define the filtration of a graph $G$, we adopt a non-negative scale parameter $\epsilon$, which is incrementally increased from 0. Suppose that the computed filter values $a_i$ are given in an ascending order $0 = a_0 < a_1 < a_2 < \cdots < a_N$, where $a_i \in \{f(\{v\}) : \{v\} \in G\}$. Upon reaching $\epsilon = a_1$, we construct $G^{f,1}$ from $G^{f,0} = \emptyset$ by adding the node $v_1$. When $\epsilon$ subsequently reaches at $a_2$, we extend $G^{f,1}$ to $G^{f,2}$ by adding the node $v_2$ and the edge connecting $v_2$ and the nodes in previous subgraph, i.e., $G^{f,1}$, if the edge exists. By repeating this process, we systemically define the sublevel set filtration induced by $f$ as:

$$G^{f,i} = \{\sigma \in G : \max_{v \in \sigma} f(v) \leq a_i\} = f^{-1}([0, a_i])$$
$$\emptyset = G^{f,0} \subseteq G^{f,1} \subseteq G^{f,2} \subseteq \cdots \subseteq G^{f,N-1} \subseteq G^{f,N} = G \tag{3}$$

for $0 \leq i \leq N$.

**Persistent Homology.** The homological features can be extended via tracking the filtration of $G$. Filtration leads to the notion of persistent homology group, $H_k^{(i,j)} = \ker \partial_k^i / (\mathrm{im} \partial_{k+1}^j \cap \ker \partial_k^i)$, for $1 \leq i \leq j \leq N$. By monitoring the (de)formation of homological features in each $G^{f,i}$ along the filtration, we can obtain their homological relevance, i.e., how long each homological features persist. Specifically, if a homological feature, e.g., a connected component or a loop, first appears at $G^{f,i}$, we define the *birth* of that homological feature as being at $i$. Likewise, the *death* of a homological feature is defined as $j$ if it disappears at $G^{f,j}$. The rank of persistent homology groups, $\mathrm{rank}(H_k^{(i,j)})$, represents the number of homological features which persist from $G^{f,i}$ to $G^{f,j}$ in $k$-dimension.

**Representation of Persistent Homology.** Suppose that a homological feature is born at $G^{f,i}$ and dies at $G^{f,j}$, i.e., that it persists from $i$ to $j$. It can be denoted as a tuple of birth and death pair, i.e., $(i, j)$, known as *persistence barcode*. By considering each $i$ and $j$ as coordinates and plotting the barcodes $(i, j)$ on the $\mathbb{R}^2$ plane, we can obtain a *persistence diagram* $\mathcal{D}_G$ of the graph $G$:

$$\mathcal{D}_G = \{(b, d) : (b, d) \text{ is persistence barcode of } G\} \subseteq \mathbb{R}^2. \tag{4}$$

Note that a persistence diagram $\mathcal{D}_G$ can be obtained separately with respect to the dimension of the barcodes, i.e., the dimension of the homological feature a barcode encodes. For every $n$ points $(b, d) \in \mathcal{D}_G$, i.e., $|\mathcal{D}_G| = n$, we associate a piece-wise linear function $f_{(b,d)} : \mathbb{R} \to [0, \infty)$, which is defined as :

$$f_{(b,d)}(x) = \begin{cases} 0 & \text{if} \quad x \notin (b, d], \\ x - b & \text{if} \quad x \in \left(b, \frac{b+d}{2}\right], \\ d - x & \text{if} \quad x \in \left(\frac{b+d}{2}, d\right). \end{cases} \tag{5}$$

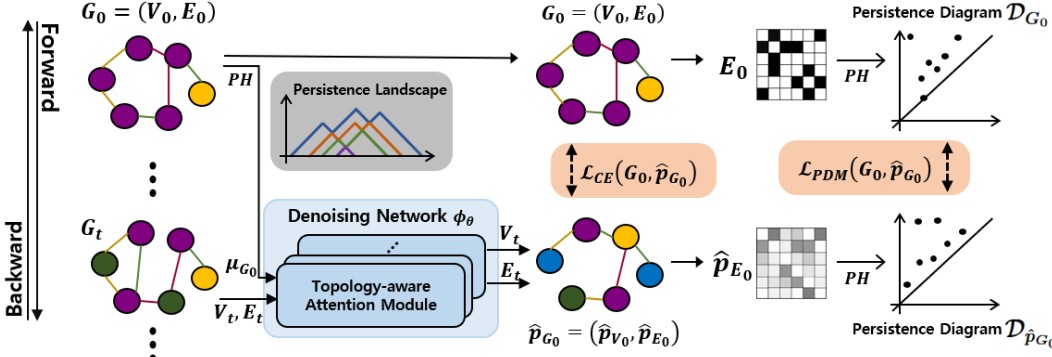

Figure 1: Training of the denoising network $\phi_\theta (G_t, \mu_{G_0})$. This network takes a noisy graph $G_t$ and the embedding $\mu_{G_0}$ obtained from persistence landscape of the original graph $G_0$ as an input, and output the probability vector of nodes and edges, $\hat{p}_{V_0}$ and $\hat{p}_{E_0}$, to predict $G_0$. During training, we utilize these predictions in two ways: **1)** cross-entropy loss $\mathcal{L}_{CE}$ over all nodes and edges, and **2)** Persistence Diagram Matching loss $\mathcal{L}_{PDM}$ which computes the discrepancy between persistence diagrams of $G_0$ and $\hat{p}_{G_0}$.

*Persistence landscape* (Bubenik, 2015) of a persistence diagram $\mathcal{D}_G$ can be established as a sequence of functions $\lambda_l : \mathbb{R} \rightarrow [0, \infty)$ for $l \in \mathbb{N}$, where $\lambda_l(x)$ denotes the $l^{\text{th}}$ largest value of the functions $f_{(b_i, d_i)}(x)$, for $i \leq n$. We obtain $s$ points by dividing the domain of the function $\lambda_l(x)$ where $\lambda_l(x) \geq 0$ into $(s+1)$ equal sub-intervals. With chosen $L$ and $S \in \mathbb{R}$, we can obtain a homological feature vector $\mu_G \in \mathbb{R}^{L \cdot S}$ for a given graph $G$ as follows:

$$\mu_G^l = [\lambda_l(x_1), \ldots, \lambda_l(x_s), \ldots, \lambda_l(x_S)] \in \mathbb{R}^S$$
$$\mu_G = [\mu_G^1, \ldots, \mu_G^l, \ldots, \mu_G^L] \in \mathbb{R}^{L \cdot S}, \tag{6}$$

where $1 \leq l \leq L$, and $1 \leq s \leq S$.

These representations, the persistence diagram $\mathcal{D}_G$ and the homological feature vector $\mu_G$, encode the entire information about persistent homology of a given graph (Edelsbrunner & Harer, 2022; Bubenik, 2015).

## 4 TAGG: TOPOLOGY-AWARE GRAPH GENERATION

Unlike conventional diffusion-based graph generation, which utilizes Gaussian noise in continuous space, we perform the diffusion process in discrete space to preserve the sparsity of the graph for every diffusion time steps. We follow the settings in Vignac et al. (2023), who successfully expands the method in Austin et al. (2021) for generating graphs with categorical node and edge attributes by treating each node and edge as a categorical random variable.

Let $G_0 = (V_0, E_0)$ be the original graph with $N$ nodes, where $V_0 \in \mathbb{R}^{N \times F_V}$ and $E_0 \in \mathbb{R}^{N \times N \times F_E}$ are the node and edge matrices with $F_V$ and $F_E$ attributes, respectively. At time step $t$, we denote the attribute of the $i$-th node $v^i$ as a one-hot vector $v_t^i \in \mathbb{R}^{F_V}$ and denote the attribute of the edge $e^{i,j}$ between $v^i$ and $v^j$ as a one-hot vector $e_t^{i,j} \in \mathbb{R}^{F_E}$. In this way, the elements of $V_t$ and $E_t$ are given by each one-hot vectors.

### 4.1 FORWARD PROCESS

Considering node and edge attributes as one-hot vectors, we follow the settings of Austin et al. (2021) and Vignac et al. (2023) to define the forward and reverse process of diffusion acting on the node and edge attributes. We denote the forward diffusion process of each time step to impose noise as transition matrices $Q_t$, where $t = 1, 2, \ldots, T$, and each element of the matrices, $[Q_t]_{\eta, \xi}$, represents the probability that state $\eta$ changes to state $\xi$ as the time step changes from $t-1$ to $t$, i.e., $[Q_t^V]_{\eta^V, \xi^V} = q(v^t = \xi^V \mid v^{t-1} = \eta^V)$ and $[Q_t^E]_{\eta^E, \xi^E} = q(e^t = \xi^E \mid e^{t-1} = \eta^E)$.

From time step $t-1$, a noised graph $G_t$ can be obtained by sampling the type of nodes and edges from the categorical distribution after transition, which is derived as:

$$q\left(G_t \mid G_{t-1}\right) = \left(V_{t-1}Q_t^V, E_{t-1}Q_t^E\right)$$

$$= \left(\begin{bmatrix} \mathrm{Cat}\left(v_t^1; v_{t-1}^1 Q_t^V\right) \\ \vdots \\ \mathrm{Cat}\left(v_t^N; v_{t-1}^N Q_t^V\right) \end{bmatrix}, \begin{bmatrix} \mathrm{Cat}\left(e_t^{1,1}; e_{t-1}^{1,1}Q_t^E\right) & \cdots & \mathrm{Cat}\left(e_t^{1,N}; e_{t-1}^{1,N}Q_t^E\right) \\ \vdots & \ddots & \vdots \\ \mathrm{Cat}\left(e_t^{N,1}; e_{t-1}^{N,1}Q_t^E\right) & \cdots & \mathrm{Cat}\left(e_t^{N,N}; e_{t-1}^{N,N}Q_t^E\right) \end{bmatrix}\right), \quad (7)$$

where $\mathrm{Cat}(z; \omega)$ denotes the categorical distribution over one-hot row vector $z$ with a probability vector $\omega$, and the dimensions of $\omega$ are $F_V$ and $F_E$ for node and edge, respectively. Specifically, the transition matrix $Q_t^V$ is determined by the dimension of node categories, i.e., $Q_t^V = (1 - \beta_t)\boldsymbol{I} + (\beta_t/F_V)\boldsymbol{J}_{F_V}$ where $\boldsymbol{I} \in \mathbb{R}^{F_V \times F_V}$ is an identity matrix and $\boldsymbol{J}_{F_V} \in \mathbb{R}^{F_V \times F_V}$ is a matrix of ones, and $\beta_t$ is a real value in range $[0, 1]$. The transition matrix for edge $Q_t^E$ is determined in the same manner.

Assuming Markovian property of the process, we can derive the transition matrix from time $0$ to time $t$ by simply multiplying each transition matrices: $\bar{Q}_t^V = Q_1^V \cdot Q_2^V \cdots Q_t^V$, and $\bar{Q}_t^E = Q_1^E \cdot Q_2^E \cdots Q_t^E$. Then, similar to Eq. (7), the noised graph $G_t$ can also be obtained from time $0$ by sampling from $q\left(G_t \mid G_0\right) = \left(V_0 \bar{Q}_t^V, E_0 \bar{Q}_t^E\right)$.

## 4.2 TOPOLOGY-AWARE DENOISING NETWORK

In this section, we introduce a topology-aware graph denoising network $\phi_\theta$ parametrized by $\theta$, which estimates the probability vector of the nodes and edges of the original graph $G_0$. In addition to the noisy graph $G_t$, we leverage the computed $\mu_{G_0}$, a vectorized representation of homological information of $G_0$ obtained via Eq. (6), as an input to the denoising network $\phi_\theta$ to retain the topological structure of the original graph during estimation.

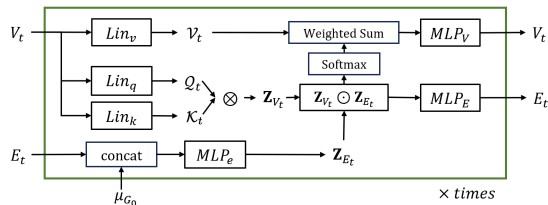

Figure 2: Topology-aware attention module of TAGG. $\otimes$ and $\odot$ denotes the outer product and element-wise product.

Specifically, we utilize the Graph Transformer Network (Dwivedi & Bresson, 2020) to estimate the categorical probability vector of the original graph for each node and edge, i.e., $\hat{p}_{G_0} = (\hat{p}_{V_0}, \hat{p}_{E_0})$, given a noisy graph $G_t$. While traditional graph transformer models rely on the self-attention modules on the node and edge embeddings (Dwivedi & Bresson, 2020; Vignac et al., 2023), we introduce a novel edge-homology embedding that incorporates the homological feature $\mu_{G_0}$ of the original graph to the attention module in the denoising network $\phi_\theta(G_t, \mu_{G_0})$.

The attention mechanism of our topology-aware denoising network $\phi_\theta$ is defined as follows:

$$\begin{aligned} \mathcal{Q}_t = \mathcal{N}_q(V_t) \qquad & \mathcal{K}_t = \mathcal{N}_k(V_t) \qquad && \mathcal{V}_t = \mathcal{N}_v(V_t) \\ \mathbf{Z}_{V_t} = \mathcal{Q}_t \otimes \mathcal{K}_t \qquad & \mathbf{Z}_{E_t} = \mathcal{N}_e(E_t, \mu_{G_0}) && \\ V_t = \mathcal{N}_V(\mathbf{Z}_{V_t}, \mathbf{Z}_{E_t}, \mathcal{V}_t) \qquad & E_t = \mathcal{N}_E(\mathbf{Z}_{V_t}, \mathbf{Z}_{E_t}), && \end{aligned} \qquad (8)$$

where $\mathcal{N}(\cdot)$ with different subscripts denotes different neural networks, and $\otimes$ and $\odot$ denote the outer and element-wise product, respectively. Also, $\mathcal{Q}_t$, $\mathcal{K}_t$, and $\mathcal{V}_t$ are the embedding vectors representing the query, key, and value, each obtained by $\mathcal{N}_q(\cdot), \mathcal{N}_k(\cdot)$ and $\mathcal{N}_v(\cdot)$. The edge-homology embedding $\mathbf{Z}_{E_t}$, obtained by incorporating the edge embeddings $E_t$ and the homological feature $\mu_{G_0}$ using $\mathcal{N}_e(\cdot)$, as well as the intermediate attention score $\mathbf{Z}_{V_t}$ and the value embedding $\mathcal{V}_t$ are then combined and passed through $\mathcal{N}_V(\cdot)$ to produce a new topology-aware node embedding. Likewise, $\mathcal{N}_E(\cdot)$ produces a topology-aware edge embedding by utilizing $\mathbf{Z}_{V_t}$ and $\mathbf{Z}_{E_t}$. By repeating the process in Eq. (8), we consequently obtain the final estimations $\hat{p}_{V_0}$ and $\hat{p}_{E_0}$. Fig. 2 further demonstrates the topology-aware attention module of TAGG.

Since $\mu_{G_0}$ encodes the homological information by tracking every subgraph in the filtration of $G_0$ (from Eq. (3)), it contains a rich amount of structural information that is challenging for the network to capture from the final subgraph of the filtration, i.e., the original graph. By incorporating $\mu_{G_0}$, the attention mechanism of the denoising network can estimate topology-aware node and edge embeddings, as described in Eq. (8), thereby enabling the final estimation of the topology-aware probability vector $\hat{p}_{G_0}$. Consequently, the edge-homology embeddings bolster the denoising network $\phi_\theta$ in generating more realistic graphs.

Moreover, it is noteworthy that the homological feature vector $\mu_{G_0}$ can be pre-computed during the data preprocessing step, thereby minimizing the need for additional computation for training. We empirically demonstrate in Sec. 5 that the homological feature vector $\mu_{G_0}$ helps the model to better learn the original distribution of nodes and edges.

### 4.3 TRAINING OBJECTIVE FOR TOPOLOGY PRESERVING GRAPH GENERATION

To produce accurate estimation of the probability vector $\hat{p}_{G_0}$, we optimize the denoising network $\phi_\theta$ with two loss terms: 1) Persistence Diagram Matching loss, which aligns the homological features of the generated graphs, and 2) Cross Entropy loss, which ensures the node and edge attributes of the generated graphs to closely resemble those of the original graphs.

We first introduce our **Persistence Diagram Matching loss** $\mathcal{L}_{PDM}$. As discussed in Sec. 3, persistence diagrams hold comprehensive homological information of graphs obtained via persistent homology. In order to let the generated graphs to resemble the homological features of the original graphs, which is our main contribution, we aim to minimize the discrepancy between the persistence diagrams of the original and the generated graph.

Given the original adjacency $G_0$ and the estimate $\hat{p}_{G_0}$, the persistence diagrams $\mathcal{D}_{G_0}$ and $\mathcal{D}_{\hat{p}_{G_0}}$ can be computed, as defined in Eq. (4). Considering the diagrams as a distribution (Lacombe et al., 2018), we calculate the discrepancy between the two distributions via 1-Wasserstein distance $W_1(\cdot)$ as:

$$\mathcal{L}_{PDM}(G_0, \hat{p}_{G_0}) = W_1(\mathcal{D}_{G_0}, \mathcal{D}_{\hat{p}_{G_0}}) = \inf_\pi \left( \sum_{\boldsymbol{x} \in \mathcal{D}_{G_0}} ||\boldsymbol{x} - \pi(\boldsymbol{x})|| \right) \tag{9}$$

where $\boldsymbol{x} \in \mathcal{D}_{G_0}$, and any bijection $\pi : \mathcal{D}_{G_0} \to \mathcal{D}_{\hat{p}_{G_0}}$.

Note that the bijection $\pi$ between two persistence diagrams holds the following two challenges: 1) if the persistence diagrams have different number of points, which is true in most cases, bijection does not exist, and 2) the matching between points from two different persistence diagrams may be misleading, i.e., the two matched points may hold homological features of different dimension, such as matching a connected component to a loop. Hence, following the common approach in TDA (Kerber et al., 2017), we pad the diagrams with points on the diagonal to ensure a proper matching between points of the two persistence diagrams. Also, as described in Sec. 3, a persistence diagram of each dimension can be acquired separately by plotting the barcodes of each dimension. Therefore, the PDM loss is computed over $\mathcal{D}_{G_0}$ and $\mathcal{D}_{\hat{p}_{G_0}}$ of the same dimension, with the bijection guaranteed to match points with homological features of the same dimension. The final $\mathcal{L}_{PDM}$ is determined by summing the distances across all dimensions.

In addition, we guide the probability vectors from $\phi_\theta$ to approximate the ground truth attributes of clean $G_0$. This is conventionally done by minimizing the following loss over all nodes and edges:

$$\mathcal{L}_{CE}(G_0, \hat{p}_{G_0}) = \mathcal{L}_{CE}^V(G_0, \hat{p}_{G_0}) + \alpha_1 \mathcal{L}_{CE}^E(G_0, \hat{p}_{G_0})$$
$$= \sum_{1 \leq i \leq N} CE\left(v^i, \hat{p}_{v_t^i}\right) + \alpha_1 \sum_{1 \leq i,j \leq N} CE\left(e^{i,j}, \hat{p}_{e_t^{i,j}}\right), \tag{10}$$

where $CE(\cdot)$ denotes the cross-entropy function, and $\alpha_1 \in (0, 1]$ denotes a real value.

The final training objective linearly combines the two losses above as:
$$\mathcal{L}_{final} = \mathcal{L}_{CE}(G_0, \hat{p}_{G_0}) + \alpha_2 \mathcal{L}_{PDM}(G_0, \hat{p}_{G_0}), \tag{11}$$
for a real value $\alpha_2 \in (0, 1]$. During the training process, $\mathcal{L}_{PDM}$ helps the network to learn the topological structure of the original graphs. The overall training framework is shown in Fig. 1 and the effect of $\mathcal{L}_{PDM}$ is validated in Sec. 5. For further details of TAGG, refer to Appendix B.

### 4.4 REVERSE PROCESS

After optimizing the denoising model $\phi_\theta$, we utilize the reverse process in Vignac et al. (2023) to generate new graphs. By iteratively estimating the denoised graph $\hat{p}_{G_0}$ given a noisy graph $G_t$ and imposing noise to the estimated graphs $\hat{p}_{G_0}$ by $q(G_{t-1} \mid G_0)$ from $t = T$ to $t = 1$, we can sample a new graph. Note that, unlike in the training step where each homological feature vector $\mu_{G_0}$ is derived from its corresponding original graph $G_0$, the matching of $\mu_{G_0}$ to its original graph $G_0$ cannot be defined in the reverse process, thus requiring homological feature vectors based on graphs from the training dataset. Hence, we utilize the averaged homological feature vector, denoted as $\mu_{G'} \in \mathbb{R}^{L \cdot S}$, whose average operation is performed over all training graphs.

# 5 EXPERIMENTS

## 5.1 DATASET AND EXPERIMENTAL SETTINGS

**Benchmark Graph Datasets.** To obtain a coherent analysis of graph generation performance, we adopt three conventional benchmark datasets of real and synthetic graphs: (1) Community-small: 200 synthetic graphs with $12 \leq |V| \leq 20$ generated from a stochastic block model with two communities, (2) Ego-small: 200 small sub-graphs of the Citeseer network dataset (Sen et al., 2008) with $4 \leq |V| \leq 18$, and (3) ENZYMES: 600 protein tertiary structures of the enzymes in graphs from the BRENDA database (Schomburg et al., 2004).

**ADNI.** To validate practicability of TAGG, we use brain connectivity from Alzheimer's Disease Neuroimaging Initiative (ADNI). In house tractography pipeline was applied to Diffusion Weighted Imaging (DWI) of healthy subjects from ADNI adhering to the Destrieux atlas (Destrieux et al., 2010) with 160 regions of interest (ROIs) comprising 148 cortical and 12 sub-cortical regions. The dataset is composed of $N = 844$ undirected weighted graphs, i.e., structural brain connectivity, where the edge weights represent the number of fiber tracts connecting different ROIs. The edges were thresholded by removing those below 5%p of the maximum edge weights to obtain sparsity.

**Baselines and Quantitative Metrics.** We used the following one-shot deep generative methods as baselines: EDP-GNN (Niu et al., 2020), GDSS (Jo et al., 2022), DiGress (Vignac et al., 2023), and the one-shot version of LocalPPGN (Bergmeister et al., 2024). In addition, a conventional auto-regressive generation method, i.e., GraphRNN (You et al., 2018b), is used for comparison. The (dis)similarity between distributions of graph statistics on the same number of generated and test graphs were computed using the maximum mean discrepancy (MMD) (Gretton et al., 2012). Specifically, we compared the distributions of degree, clustering coefficient, and the number of occurrences of orbits with 4 nodes, as in Niu et al. (2020) and Jo et al. (2022). For consistency, we adhered to the train/test split reference from Jo et al. (2022) and performed three replicate experiments to report averaged performance on all models.

**Visualization of Homological Assessment.** We utilized two distinct measures to assess the homological (dis)similarity between the test and generated graphs: 1) Automatic Topologically-Oriented Learning (ATOL) (Royer et al., 2021), and 2) Mean Landscape (Bubenik, 2015). These methods directly encode the homological features from the persistence diagrams into vectors of desired dimensions, thus facilitating a topology-aware assessment of the generated graphs. We further visualized the generated graphs for qualitative comparison.

## 5.2 QUANTITATIVE ANALYSIS

**Results.** The comparison between the baselines and the proposed method is shown in Tab. 1. Note that although the models for the best and the second best performances vary in each metric, the proposed method steadily shows highly promising performance, especially in the averaged value of the three MMD metrics. Specifically, we observed that Vignac et al. (2023), the referenced discrete diffusion based graph generation method, did not perform well on small-scale graphs, i.e., Community-small and Ego-small, whereas TAGG showed better performance across all metrics.

Despite the superior performance, it is hard to observe how the proposed topology-aware learning framework effects the training of a generation model. Hence, we provide two ablation studies to empirically show their effect on the distribution learning of graphs in the following.

Table 1: Quantitative comparison with baseline models on synthetic and real graph datasets. The best and second best results are highlighted in **bold** and underline, respectively. The values are the averaged performance of 3 different runs.

| Method | ADNI | | | | ENZYMES | | | | Community-small | | | | Ego-small | | | |
| --- | --- | --- | --- | --- | --- | --- | --- | --- | --- | --- | --- | --- | --- | --- | --- | --- |
| | Real, $|V| = 160$ | | | | Real, $10 \leq |V| \leq 125$ | | | | Synthetic, $12 \leq |V| \leq 20$ | | | | Real, $4 \leq |V| \leq 18$ | | | |
| | Deg.↓ | Clus.↓ | Orbit↓ | Avg.↓ | Deg.↓ | Clus.↓ | Orbit↓ | Avg.↓ | Deg.↓ | Clus.↓ | Orbit↓ | Avg.↓ | Deg.↓ | Clus.↓ | Orbit↓ | Avg.↓ |
| GraphRNN (You et al., 2018b) | 1.392 | 0.916 | **0.153** | 0.820 | 0.161 | 0.942 | 0.112 | 0.405 | 0.183 | 0.182 | 0.113 | 0.159 | 0.069 | 0.090 | 0.052 | 0.071 |
| EDP-GNN (Niu et al., 2020) | 1.063 | 1.430 | 0.626 | 1.039 | 0.052 | 0.895 | 0.474 | 0.474 | 0.056 | **0.038** | 0.069 | 0.054 | 0.029 | **0.046** | 0.008 | 0.028 |
| GDSS (Jo et al., 2022) | 0.949 | 1.104 | 0.165 | 0.739 | 0.314 | 0.506 | 0.084 | 0.301 | **0.033** | 0.112 | **0.009** | 0.051 | 0.045 | 0.076 | 0.008 | 0.043 |
| DiGress (Vignac et al., 2023) | 0.504 | 1.168 | 0.379 | 0.683 | 0.023 | 0.051 | 0.205 | 0.093 | 0.089 | 0.091 | 0.049 | 0.076 | 0.026 | 0.090 | 0.023 | 0.046 |
| LocalPPGN (Bergmeister et al., 2024) | 0.777 | **0.149** | 0.954 | 0.627 | 0.037 | 0.068 | **0.048** | 0.051 | 0.034 | 0.218 | 0.018 | 0.090 | 0.014 | 0.091 | **0.006** | 0.037 |
| TAGG | **0.213** | 0.841 | 0.176 | **0.410** | **0.012** | **0.046** | 0.116 | 0.058 | 0.050 | 0.064 | 0.016 | 0.043 | **0.001** | 0.051 | 0.015 | **0.023** |

**Ablation study on $\mu_{G_0}$ (Homological feature).** We conducted an ablation study to evaluate the impact of the homological feature $\mu_{G_0}$. In Tab. 2, we show that the feature $\mu_{G_0}$ introduced in Sec. 4.2, enhances the denoising network to generate realistic graphs by providing the underlying global structural information. Note that Tab. 2 shows performance gain on all metrics when utilizing $\mu_{G_0}$, with the exception of the clustering coefficient and the averaged MMD score for Ego-small dataset, which consists of the smallest graphs with an average of 6.41 nodes. This may be attributed to the small size of the graphs, which allows the denoising network to sufficiently capture the global structural information within a few hops.

Table 2: Ablation study on $\mu_{G_0}$. Gain refers to the performance gain obtained by adding $\mu_{G_0}$ to the denoising network, where lower score means better performance with lower MMD discrepancy. $|\bar{V}|$ denotes the average number of nodes on each dataset.

| Metric | ADNI $|\bar{V}| = 160, (|V| = 160)$ | | | ENZYMES $|\bar{V}| = 32.63, (10 \leq |V| \leq 125)$ | | | Community-small $|\bar{V}| = 15.28, (12 \leq |V| \leq 20)$ | | | Ego-small $|\bar{V}| = 6.41, (4 \leq |V| \leq 18)$ | | | Avg. Gain |
|---|---|---|---|---|---|---|---|---|---|---|---|---|---|
| | w/o $\mu_{G_0}$ | with $\mu_{G_0}$ | Gain | w/o $\mu_{G_0}$ | with $\mu_{G_0}$ | Gain | w/o $\mu_{G_0}$ | with $\mu_{G_0}$ | Gain | w/o $\mu_{G_0}$ | with $\mu_{G_0}$ | Gain | |
| Deg.↓ | 0.399 | 0.213 | -0.186 | 0.017 | 0.012 | -0.005 | 0.106 | 0.050 | -0.056 | 0.007 | 0.001 | -0.006 | **-0.063** |
| Clus.↓ | 1.006 | 0.841 | -0.165 | 0.049 | 0.046 | -0.003 | 0.123 | 0.064 | -0.059 | 0.039 | 0.051 | 0.015 | **-0.053** |
| Orbit↓ | 0.267 | 0.176 | -0.091 | 0.146 | 0.116 | -0.030 | 0.020 | 0.016 | -0.004 | 0.021 | 0.015 | -0.006 | **-0.033** |
| Avg.↓ | 0.557 | 0.410 | -0.147 | 0.071 | 0.058 | -0.013 | 0.083 | 0.043 | -0.040 | 0.022 | 0.023 | 0.001 | **-0.050** |

**Ablation study on $\mathcal{L}_{PDM}$.** To evaluate the effectiveness of our PDM loss $\mathcal{L}_{PDM}$, we also provide an ablation study in Tab. 3. Similar to the method explained in Sec. 4.3, we applied the $\mathcal{L}_{PDM}$ on the persistence diagrams of the original clean graph $G_0$ and the estimate $\hat{p}_{G_0}$ on the one-shot graph generation baselines to observe the general performance gap when utilizing $\mathcal{L}_{PDM}$. Shown in Tab. 3, we empirically demonstrate that $\mathcal{L}_{PDM}$ successfully guides the network to produce topologically reliable graphs. In almost all datasets, the average metric values improved by $1.2 \sim 2$ times for both the baselines and TAGG, highlighting the impact of $\mathcal{L}_{PDM}$.

Table 3: Ablation study on the persistence diagram matching loss $\mathcal{L}_{PDM}$. Results are the averaged performance of 3 replicates, and the superior values are given in **bold**.

| Method | ADNI | | | | ENZYMES | | | | Community-small | | | | Ego-small | | | |
|---|---|---|---|---|---|---|---|---|---|---|---|---|---|---|---|---|
| | Deg.↓ | Clus.↓ | Orbit↓ | Avg.↓ | Deg.↓ | Clus.↓ | Orbit↓ | Avg.↓ | Deg.↓ | Clus.↓ | Orbit↓ | Avg.↓ | Deg.↓ | Clus.↓ | Orbit↓ | Avg.↓ |
| EDP-GNN | 1.063 | 1.430 | 0.626 | 1.039 | **0.052** | 0.895 | 0.474 | 0.474 | 0.056 | **0.038** | 0.069 | 0.054 | 0.029 | 0.046 | 0.008 | 0.028 |
| EDP-GNN+$\mathcal{L}_{pdm}$ | **1.011** | **0.854** | **0.600** | **0.822** | 0.134 | **0.729** | **0.143** | **0.335** | **0.041** | 0.041 | **0.015** | **0.032** | **0.024** | **0.041** | **0.007** | **0.024** |
| GDSS | 0.949 | 1.104 | **0.165** | 0.739 | 0.314 | 0.506 | 0.084 | 0.301 | 0.033 | 0.112 | 0.009 | 0.051 | 0.045 | 0.076 | **0.008** | 0.043 |
| GDSS+$\mathcal{L}_{pdm}$ | **0.403** | **0.675** | 0.539 | **0.539** | **0.127** | 0.529 | **0.058** | **0.238** | **0.026** | **0.096** | **0.005** | **0.043** | **0.040** | **0.059** | 0.010 | **0.036** |
| DiGress | 0.504 | 1.168 | 0.379 | 0.683 | 0.023 | 0.051 | 0.205 | 0.093 | **0.089** | **0.091** | 0.049 | **0.076** | 0.026 | 0.090 | 0.023 | 0.046 |
| DiGress+$\mathcal{L}_{pdm}$ | **0.399** | **1.006** | **0.267** | **0.557** | **0.017** | **0.049** | 0.146 | **0.071** | 0.106 | 0.123 | **0.020** | 0.083 | **0.007** | **0.039** | **0.021** | **0.022** |
| LocalPPGN | 0.777 | 0.149 | 0.954 | 0.627 | 0.037 | 0.068 | 0.048 | 0.051 | 0.034 | 0.218 | **0.018** | 0.090 | 0.014 | 0.091 | **0.006** | 0.037 |
| LocalPPGN+$\mathcal{L}_{pdm}$ | **0.374** | **0.120** | **0.695** | **0.396** | **0.025** | **0.061** | **0.031** | **0.039** | **0.030** | **0.154** | 0.019 | **0.068** | **0.012** | **0.081** | 0.009 | **0.034** |
| TAGG(w/o $\mathcal{L}_{pdm}$) | 0.379 | 1.263 | 0.247 | 0.630 | **0.010** | 0.049 | 0.148 | 0.069 | **0.047** | 0.079 | 0.028 | 0.051 | 0.005 | 0.060 | 0.017 | 0.027 |
| TAGG | **0.213** | **0.841** | **0.176** | **0.410** | 0.012 | **0.046** | **0.116** | **0.058** | 0.050 | **0.064** | **0.016** | **0.043** | **0.001** | **0.051** | **0.015** | **0.023** |

==Moreover, the improvements in clustering and orbit metrics underscore the effectiveness of $\mu_{G_0}$ and $\mathcal{L}_{PDM}$ in preserving essential topological structures. The improved clustering metrics demonstrate TAGG's ability to capture local connectivity patterns, while the improved orbit metrics reflect its capacity to preserve structural patterns in each node's local subgraphs. These findings validate the significant contribution of the proposed topology-aware framework in generating high-fidelity graphs.==

## 5.3 QUALITATIVE ANALYSIS

**Visualization of generated graphs.** We qualitatively compare TAGG with other baselines via visualization. As seen in Fig. 3 and 4, TAGG produces more realistic graphs that closely resemble test graphs from various datasets. More visualization of the generated graphs of TAGG can be found in Appendix A. Baseline models (Niu et al., 2020; Jo et al., 2022; Vignac et al., 2023; Bergmeister et al., 2024) often failed to capture the nuanced patterns present in benchmark datasets. Particularly in Fig. 3, TAGG better represent critical characteristics compared to baselines, i.e., sparse inter-hemisphere connections and symmetry of hemispheres. Although the brain networks generated from the other discrete diffusion method, i.e., Vignac et al. (2023), also exhibit high quality, they failed to capture the inter-hemisphere connection, a subtle but critical structural information of a brain network. Also in a topological perspective, the existence of inter-hemisphere connection determines the number of connected components, and the topology-aware generation of TAGG is capable of capturing such topological structures. This demonstrates the effectiveness of topology-aware learning in capturing both global and detailed structural properties of a graph.

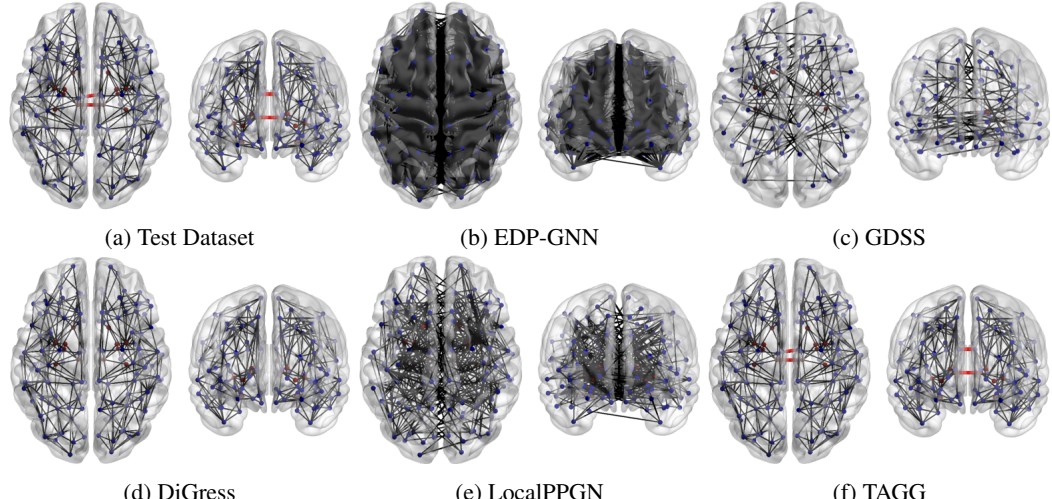

Figure 3: Visualization of the averaged brain network of 50 samples from (a) test dataset, (b) EDP-GNN, (c) GDSS, (d) DiGress, (e) LocalPPGN, and (f) TAGG. The inter-hemisphere connection of (a) and (f) are colored in red to highlight the difference between the best resulting models (d) and (f). The global structure of the brain network, e.g., the sparsity and the inter-hemisphere connectivity, are well preserved using TAGG.

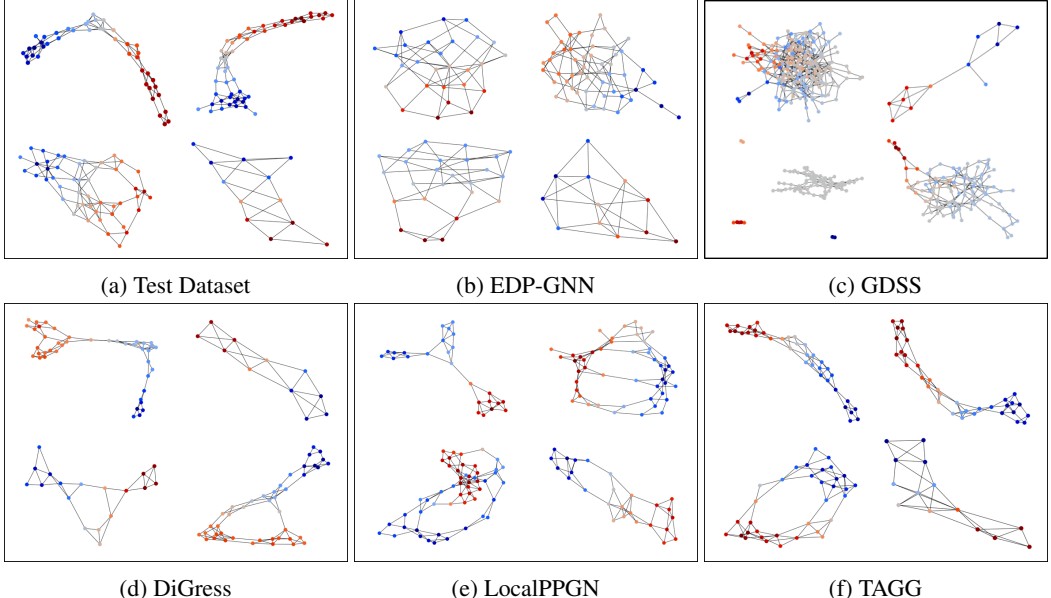

Figure 4: Visualization of the ENZYMES samples from (a) test dataset, (b) EDP-GNN, (c) GDSS, (d) DiGress, (e) LocalPPGN, and (f) TAGG. TAGG generates the most topologically equivalent graphs.

**ATOL** (Royer et al., 2021). Via ATOL, we visualize the homological feature vectors on a 2-dimensional plane to assess the impact of our topology-aware learning as test samples vs. generated samples. ATOL gets persistence diagrams derived from each graphs as inputs, and their homological feature vectors are encoded to the plottings in 2-dimensional plane. The visualization of the plotted samples in Fig. 5 thus demonstrates how closely the generated samples resemble the test samples in topological perspectives. The proximity between the pink and blue points in (e) is significantly closer compared to that observed in (a)-(d), illustrating the impact of our topology-aware generation.

**Mean Landscape** (Bubenik, 2015). Persistence diagram obtained from a graph can be transformed to a persistence landscape, where each barcode is transformed into a piecewise-linear function, as introduced in Eq. (6). Given a sequence of piecewise-linear functions $\lambda_l$ from a persistence diagram, i.e., a persistence landscape, mean landscape can be computed by taking the average of the landscapes defined as $\tilde{\lambda}(x) = \frac{1}{N} \sum_{l=1}^{N} \lambda_l(x)$, where $N$ is the number of piecewise-linear functions and $\lambda_l(x)$

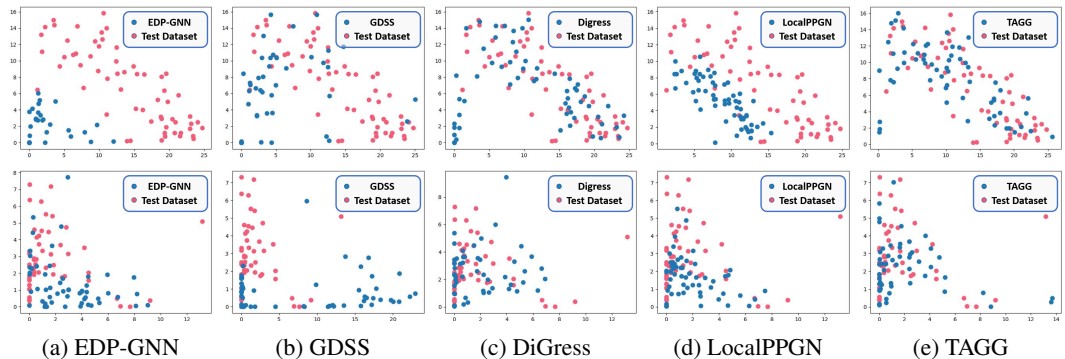

Figure 5: ATOL visualization of homological features derived from the test (Pink) and generated graphs (Blue). Top: ADNI dataset, Bottom: ENZYMES dataset. The distribution of features from a sample from TAGG exhibit the best similarity with the ground truth.

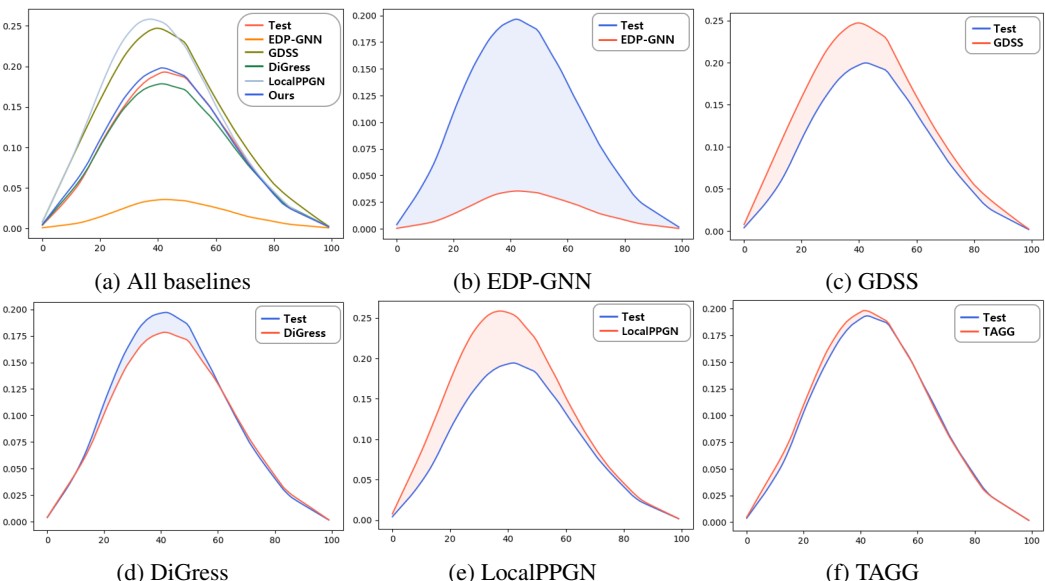

Figure 6: The averaged mean landscape (Bubenik, 2015) obtained from (a) graphs via baseline methods, and individual comparison between test graphs and (b) EDP-GNN, (c) GDSS, (d) DiGress, (e) LocalPPGN, and (f) TAGG. The mean landscape is the averaged persistence landscape of 10 piecewise-linear functions, and graphs from TAGG shows the highest resemblance to the test graphs.

is the $l^{th}$ largest value of $f_{(b_i, d_i)}(x)$ at point $x$ as described in Sec. 3. Thus, a mean landscape hold topological summary of a graph, and the similarity of two mean landscapes implies that their topological structures, e.g., connected components and loops, are analogous. To further investigate the validity of our method, Fig. 6 depicts the averaged mean landscape of the test and generated graphs on each baseline models. As can be seen in Fig. 6, TAGG significantly resembles the averaged mean landscape of the test graphs compared to all of the baseline methods. This demonstrates that TAGG generates graphs with high fidelity in the lens of graph topology, i.e., homological features.

## 6    CONCLUSION

In this study, we proposed a novel graph generation framework preserving the intricate topology of the network. Through the proposed topology-aware attention module and the Persistence Diagram Matching Loss, we achieve high generation performance while maintaining the essential topological features of the original graphs. This approach improves the fidelity of generated graphs and provides valuable insights into their structure for both synthetic and real graph datasets. Our research addresses a critical challenge in complex real-world graph generation, particularly in the context of brain networks, and pave the way for practical graph generation with topological consistency.

## REPRODUCIBILITY STATEMENT

To ensure the reproducibility of our work, we provide a detailed figure of the topology-aware attention module in Fig. 2, along with the algorithm of our general framework in Appendix B. Furthermore, we provide the experimental results, averaged across multiple runs, and the additional implementation details for TAGG and the baseline methods are provided in Appendix C. Thus, the experimental results should remain consistent across different users, and we will make the code public that produce the same result in this manuscript.

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

# A QUALITATIVE RESULTS

In this section, we provide additional generated graphs that has not been included in the main paper due to space limit. For example, Fig. 7 includes the (a) brain network and (b) ENZYMES graphs generated by GraphRNN, a baseline method of our paper. Align with the qualitative comparison results shown in Sec. 5 of the main paper, TAGG better represents critical characteristics compared to GraphRNN. Also, Fig. 8,9,10 shows additional graphs generated using TAGG to validate the consistency of the generation performance.

## A.1 GENERATED GRAPH SAMPLES VIA GRAPHRNN.

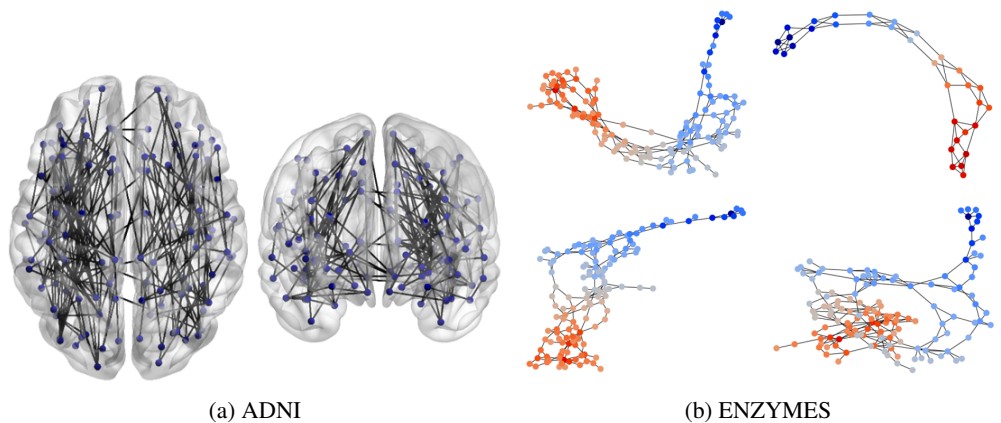

(a) ADNI                              (b) ENZYMES

Figure 7: Visualization of the generated graphs using GraphRNN; (a) Brain network from ADNI dataset and (b) ENZYMES graphs.

## A.2 GENERATED BRAIN NETWORK SAMPLES VIA TAGG.

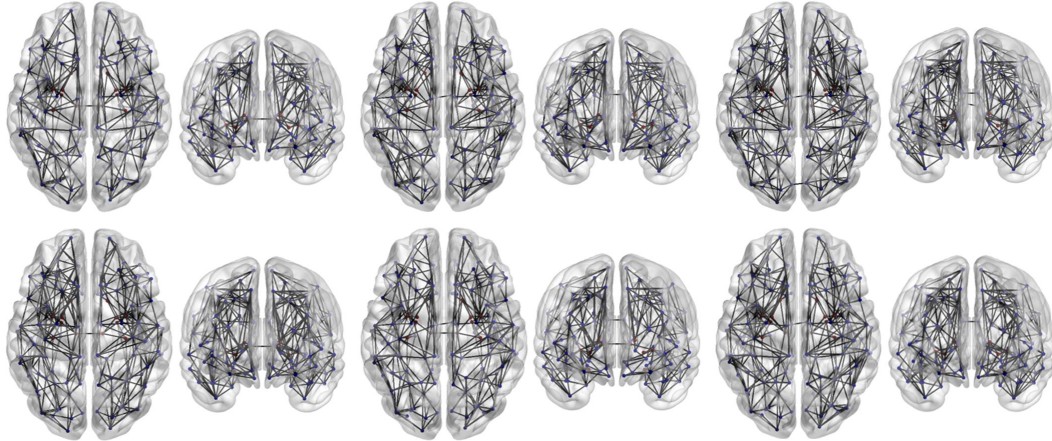

Figure 8: Visualization of the brain network generated using TAGG. TAGG successfully generates homologically reliable brain network, preserving the symmetry of brain network and the edges interconnecting left and right hemisphere.

## A.3 GENERATED SAMPLES ON BENCHMARK DATASET VIA TAGG.

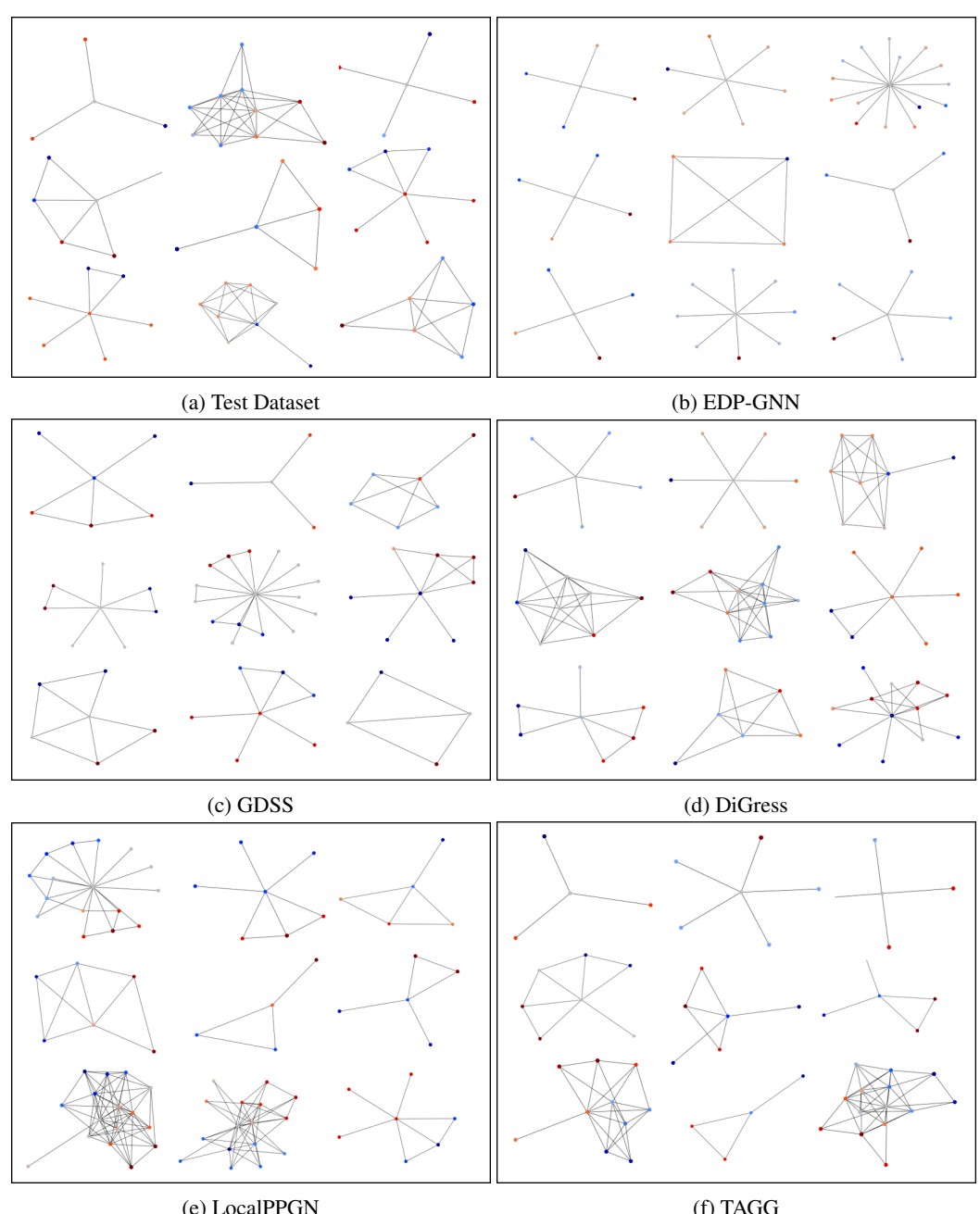

Figure 9: Visualization of the generated Ego-small graphs from (a) test dataset, (b) EDP-GNN, (c) GDSS, (d) DiGress, (e) LocalPPGN, and (f) TAGG.

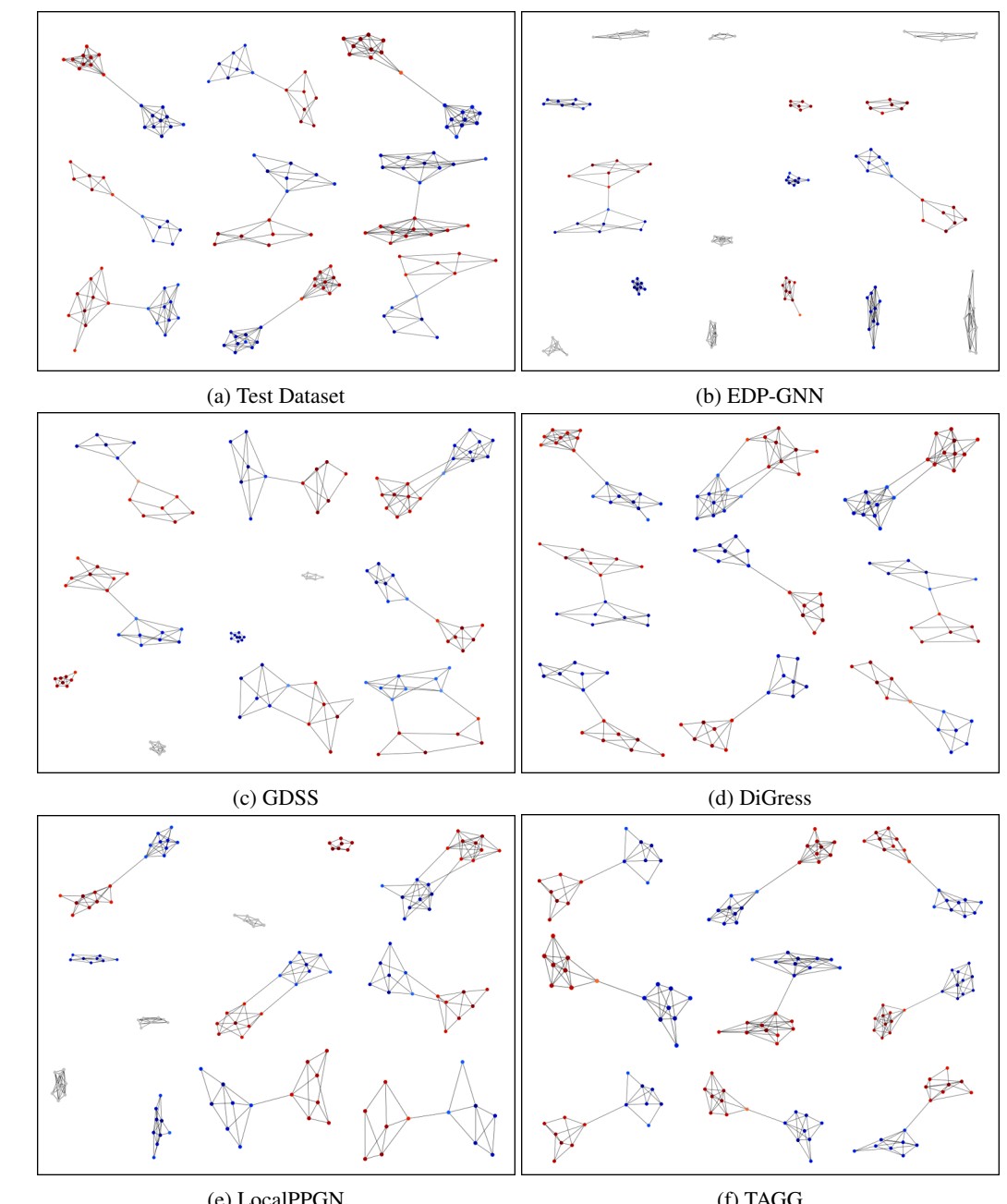

Figure 10: Visualization of the generated Community-small graphs from (a) test dataset, (b) EDP-GNN, (c) GDSS, (d) DiGress, (e) LocalPPGN, and (f) TAGG. Compared to the baselines, TAGG generates the most topologically equivalent graphs. TAGG successfully generates an edge that connects two communities.

# B DETAILS OF THE TOPOLOGY-AWARE DENOISING NETWORK

The overall scheme of TAGG is demonstrated in Algorithm 1. The homological feature vector $\mu_{G_0}$ is obtained via persistence landscape of the original graph $G_0$, which is derived from the filtration of $G_0$. Then, using the resultant $\mu_{G_0}$, the denoising network of TAGG iteratively utilize the topology-aware attention module. The homological feature $\mu_{G_0}$ enhance the attention module to estimate topology-aware node and edge embeddings, i.e., $V_t$ and $E_t$, which leads to high fidelity of generated graphs. Consequently, the denoising network outputs a probability vector $\hat{p}_{G_0} = (\hat{p}_{V_0}, \hat{p}_{E_0})$ of the original graph, which is then optimized using the Cross-Entropy and Persistence Diagram Matching loss.

---

**Algorithm 1** Overall scheme of TAGG

---

1: **Input:** Original graph $G_0 = (V_0, E_0)$, number of diffusion step $T$, hyperparameter $\alpha_1$ and $\alpha_2$.

2: **1. Obtain homological feature** $\mu_{G_0}$
3: $\mathrm{ph}_{G_0} \leftarrow \mathrm{Filtration}(G_0)$
4: Obtain persistence barcodes and persistence diagram $\mathcal{D}_{G_0}$ from $\mathrm{ph}_{G_0}$
5: $\mu_{G_0} = \mathrm{PersistenceLandscape}(\mathcal{D}_{G_0})$         ▷ Homological feature of the given graph $G_0$.

6: **2. Training TAGG**
7: **Model input:** $G_0 = (V_0, E_0)$, $\mu_{G_0}$
8: Sample $t \sim \mathcal{U}(1, 2, ..., T)$
9: Sample noisy graph $G_t = (V_t, E_t) \sim \left(V_0 \bar{Q}_t^V, E_0 \bar{Q}_t^E\right)$

10: **2-1. Estimate $\hat{p}_{G_0}$ via TAGG**
11: Given $V_t$ and $E_t$,
12: **for** number of layers **do**
13:     $\mathcal{Q}_t, \mathcal{K}_t, \mathcal{V}_t \leftarrow \mathrm{MLP}_q(V_t), \mathrm{MLP}_k(V_t), \mathrm{MLP}_v(V_t)$
14:     $\mathbf{Z}_{V_t} = \mathcal{Q}_t \otimes \mathcal{K}_t$                  ▷ Self-attention of node features.

15:     $\mathbf{Z}_{E_t} \leftarrow \mathrm{concat}(E_t, \mu_{G_0})$           ▷ Incorporate $\mu_{G_0}$ to edge embedding.
16:     $\mathbf{O}_E, \mathbf{U}_E \leftarrow \mathrm{MLP}_O(\mathbf{Z}_{E_t}), \mathrm{MLP}_U(\mathbf{Z}_{E_t})$
17:     $\mathbf{Y} \leftarrow \mathbf{Z}_{V_t} \odot \mathbf{O}_E + \mathbf{U}_E$

18:     $\mathbf{Z}'_{V_t} \leftarrow \mathrm{Softmax}(\mathbf{Y})$
19:     $V_t, E_t \leftarrow \mathrm{MLP}_V(\mathbf{Z}'_{V_t} \odot \mathcal{V}_t), \mathrm{MLP}_E(\mathbf{Y})$
20: **end for**
21: $\hat{p}_{G_0} = (\hat{p}_{V_0}, \hat{p}_{E_0}) = (\mathrm{LayerNorm}(V_t), \mathrm{LayerNorm}(E_t))$

22: **Model output:** probability of denoised graph $\hat{p}_{G_0}$

23: **2-2. Training Objective**
24: $\mathcal{L}_{\mathrm{final}} = \mathcal{L}_{\mathrm{CE}}^V(G_0, \hat{p}_{G_0}) + \alpha_1 \mathcal{L}_{\mathrm{CE}}^E(G_0, \hat{p}_{G_0}) + \alpha_2 \mathcal{L}_{PDM}(G_0, \hat{p}_{G_0})$

---

## C    IMPLEMENTATION DETAILS

We provide additional details of experiment settings used in TAGG. As explained in Sec. 4.3, the training objective of TAGG has two real valued hyperparameters $\alpha_1 \in (0, 1]$ and $\alpha_2 \in (0, 1]$, each used to control the cross-entropy loss of edges $\mathcal{L}_{\text{CE}}^E$ and the persistence diagram matching (PDM) loss $\mathcal{L}_{PDM}$, respectively. The hyperparameters $\alpha_1$ and $\alpha_2$ were chosen through a grid search of values in $\{1, 0.1, 0.01, 0.001, 0.0001\}$ on each dataset, and the settings are shown in Tab. 4. We followed the hyperparameters provided in the original papers for the baseline methods. For the datasets that were not included in the original papers, we conducted the same hyperparameter search as with TAGG to ensure a fair comparison. Additionally, after the reverse diffusion process to sample the generated graphs, we quantize the entries of the adjacency matrices using the operator $1_{x>0.5}$, resulting in a binary adjacency matrix.

Table 4: Hyperparameters of TAGG on different datasets

| Hyperparameter | ADNI | ENZYMES | Community-small | Ego-small |
|---|---|---|---|---|
| $\alpha_1$ | 1 | 1 | 0.001 | 0.01 |
| $\alpha_2$ | 0.001 | 0.0001 | 0.001 | 0.0001 |

# D   ADDITIONAL EXPERIMENT ANALYSIS

We offer a more detailed analysis of the quantitative experiments presented in Sec. 5, along with additional experiments examining the generation performance of TAGG.

## D.1   ANALYSIS OF ABLATION STUDY (SEC. 5.2).

The quantitative measure (MMD scores) of the graph generation performance with the use of homological feature vector $\mu_{G_0}$ and PDM loss $\mathcal{L}_{PDM}$ is shown in Tab. 2 and Tab. 3. The ablation study demonstrate that incorporating $\mu_{G_0}$ to the attention module and the use of PDM loss as a regularizer generally enhances performance metrics across various datasets, suggesting that TAGG effectively generates high-fidelity graphs by leveraging the proposed methods.

Moreover, the improvements in the clustering and orbit metrics underscore the impact of our method in preserving critical topological structures. Specifically, the enhanced clustering metric highlights the ability of TAGG to maintain local connectivity patterns, while the improved orbit metric—capturing the structural patterns within each node's local subgraph—shows the effectiveness of TAGG in preserving the structural roles and relationships within a graph. These results confirm that the proposed topology-aware learning framework significantly contributes generating high-fidelity graphs.

## D.2   CHOICE OF FILTER FUNCTION $f$.

The degree function was employed as the filter function in TAGG, as it is one of the most simple but common and effective ways to define the filtration of a graph (Hofer et al., 2017; 2020; Carriere et al., 2020). While the degree-based filter function may yield relatively smaller improvements in clustering and orbit metrics compared to the degree metric itself, the overall results of TAGG across all datasets demonstrate consistent improvements over other baselines in the quantitative comparison and ablation studies. Additionally, further experiments were conducted using alternative filter functions, such as the clustering coefficient and betweenness centrality. The clustering coefficient $\mathcal{C}_i$ measures the local density of connections, defined as $\mathcal{C}_i = 2e_i/k_i(k_i - 1)$, where $k_i$ is the number of neighbors of node $i$, and $e_i$ is the number of edges between those neighbors. Betweenness centrality measures how often a node lies on the shortest path between other nodes, reflecting how crucial a node is in bridging different parts of a graph. However, due to the computational complexity and differentiability, we approximate the betweenness centrality using random walks. The betweenness centrality with random walks $\mathcal{C}_{rw}$ is defined as $\mathcal{C}_{rw}(k) = \sum_{i \neq k \neq j} P_{i,j}(k)/P_{i,j}$, where $P_{i,j}$ is the total number of random walks from node $i$ to node $j$, and $P_{i,j}(k)$ is the number of random walks from node $i$ to node $j$ that pass through node $k$. As presented in the Tab. 5, changing the filter function affect the optimization process; for example, using the clustering coefficient enhances its corresponding MMD score. However, the overall impact on the generation performance remains minimal, as the new averaged MMD metrics continue to outperform the baseline methods without significant degradation.

## D.3   GENERALIZABILITY TO SMALL AND LARGE GRAPH DATASET.

Regardless of the size of the graph, the ablation studies presented in Tab. 2 and Tab. 3 demonstrate that our proposed method, which utilize the topology-aware self-attention module and PDM loss, consistently improves performance across the overall dataset. However, in the case of small graphs, its limited number of simplices results in a restricted set of topological features, which may diminish the effectiveness of our method when compared to larger graph datasets. As a result, the MMD metric results (degree, clustering, orbit) in Tab. 1 may not show significant differences. Nevertheless, the averaged MMD scores outperform all baselines on

Table 5: Comparison of filter functions across datasets. Filter functions are the degree, clustering coefficient (Clus. Coeff), and betweenness centrality (Betw. Cent.). Empty values will be updated during the Rebuttal period.

| Filter function | ADNI | | | | ENZYMES | | | |
| --- | --- | --- | --- | --- | --- | --- | --- | --- |
| | Deg.↓ | Clus.↓ | Orbit↓ | Avg.↓ | Deg.↓ | Clus.↓ | Orbit↓ | Avg.↓ |
| Degree | 0.213 | 0.841 | 0.176 | 0.410 | 0.012 | 0.046 | 0.116 | 0.058 |
| Clus. Coeff. | 0.242 | 0.802 | 0.198 | 0.414 | - | - | - | - |
| Betw. Cent. | 0.252 | 0.847 | 0.336 | 0.478 | 0.009 | 0.047 | 0.112 | 0.056 |
| Filter function | Community-small | | | | Ego-small | | | |
| | Deg.↓ | Clus.↓ | Orbit↓ | Avg.↓ | Deg.↓ | Clus.↓ | Orbit↓ | Avg.↓ |
| Degree | 0.050 | 0.064 | 0.016 | 0.043 | 0.001 | 0.051 | 0.015 | 0.023 |
| Clus. Coeff. | 0.071 | 0.057 | 0.008 | 0.045 | 0.006 | 0.040 | 0.025 | 0.023 |
| Betw. Cent. | 0.071 | 0.059 | 0.017 | 0.049 | 0.005 | 0.046 | 0.020 | 0.024 |

both the community-small and ego-small datasets, highlighting the meaningful impact of our method on generation performance of small graphs.

# E  VISUALIZATION OF HIDDEN REPRESENTATIONS

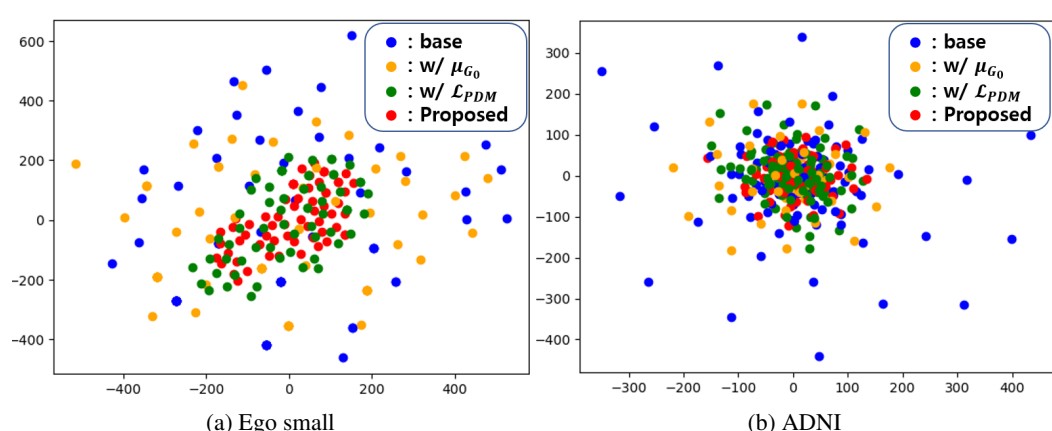

(a) Ego small                     (b) ADNI

Figure 11: t-SNE visualization of the trained features from the topology-aware attention module on (a) Ego-small and (b) ADNI dataset. Colors denote the t-SNE results from trained features under different model settings; Blue: baseline, Orange: TAGG without $\mu_{G_0}$, Green: TAGG without $\mathcal{L}_{PDM}$, and Red: TAGG.

In order to investigate the effect of the topology-aware learning framework, Fig. 11 demonstrates the t-SNE visualization of the trained features from the topology-aware attention module on different datasets. Specifically, the hidden features from the final layer of the attention module were extracted and projected onto a 2-dimensional plane using t-SNE, providing a visual representation of the trained hidden features in the latent space. To evaluate the individual contributions of the topology-aware attention module and the PDM loss of TAGG, t-SNE results were obtained using the same models from the ablation study, along with a baseline model that excludes both $\mu_{G_0}$ and $\mathcal{L}_{PDM}$. Notably, incorporating either the homological feature $\mu_{G_0}$, the PDM loss, or both consistently improved performance across all datasets. In line with the enhanced quantitative results, the visualization reveals differences in the latent space between the cases where neither method was applied and where both were utilized, indicating that the graph features were optimized into a more desirable latent space.

