# OpenReview forum: "Topology-aware Graph Diffusion Model with Persistent Homology"
_ICLR.cc/2025/Conference — ICLR 2025 Conference Withdrawn Submission_

### Official Review · Reviewer_6rDp · 2024-11-04

**Soundness:** 2
**Presentation:** 2
**Contribution:** 2
**Rating:** 3
**Confidence:** 4

**Summary:**

The paper introduces a topology-aware graph generation method called TAGG that incorporates persistent homology into diffusion models to preserve structural characteristics of graphs. The authors propose two main technical contributions: a persistence diagram matching (PDM) loss that ensures generated graphs match the topology of original graphs, and a topology-aware attention mechanism that enhances the self-attention module by incorporating homological features. The method is evaluated on various datasets, with a particular emphasis on brain network generation.

**Strengths:**

The technical execution of the work is solid and thorough. The empirical evaluation is comprehensive, with extensive comparisons against baselines and multiple visualizations. The application to brain network generation is interesting, as it addresses a real-world problem where topological features are crucial. The authors also provide detailed ablation studies.

**Weaknesses:**

The paper's primary limitation lies in its incremental nature and lack of theoretical depth. The core diffusion framework is heavily based on existing work (Vignac et al., 2023), and the integration of persistent homology, while useful, represents an incremental advance rather than a fundamental breakthrough. The topology-aware attention mechanism is essentially a straightforward modification of standard attention by incorporating topological features. And it is not clear why such topological based representations or vectorizations are differentiable and can be trained in an end-to-end generative model. The paper lacks theoretical analysis of why topology-awareness improves generation and provides no theoretical guarantees about topological preservation. The computational overhead of computing persistent homology is not adequately addressed, which could be significant for larger graphs. Additionally, the novelty is limited as the use of persistent homology in graph analysis has been explored in previous works (e.g., Hofer et al., 2020), and the paper doesn't clearly articulate how their approach fundamentally differs from these previous applications.

**Questions:**

- How does the computational complexity scale with graph size, particularly considering the overhead of computing persistent homology? How does the method perform on very large graphs, and what are the main scalability challenges?
- The topology-aware attention mechanism uses pre-computed homological features during training, but how sensitive is the performance to the choice of filtration function used to compute these features?
- Are these topology-aware attention and persistence diagram matching loss differentiable or able to be trained end-to-end?
- What new features are learned through proposed methods? Is there any way to illustrate related hidden representations?
- Could this approach be extended to dynamic graphs where topology evolves over time?

---

> ### Author Response · Authors · 2024-11-21
>
> **Regarding your weaknesses :**
>
> ---
>
> **W1) The paper's primary limitation lies in its incremental nature and lack of theoretical depth. Also, it is not clear how the proposed method can be trained in an end-to-end manner.**
>
> **A)** In our humble opinion, our contributions with respect to the core idea and concept are non-trivial although technical modification may be incremental, and this is backed up by reviews from other reviewers. The main contribution of our work lies on graph generation with topological fidelity, which is crucial in real-world applications e.g., brain network modeling. To the best of our knowledge, this is the first work to examine graph generation tasks through the lens of persistent homology, validated by comprehensive experiments conducted both qualitatively and quantitatively (which the reviewer pointed as a strength). The homological features encodes a comprehensive topological information of a graph, obtained via persistent homology, which is used to enhance the denoising network and regularizing the model for topology-aware graph generation.
>
> Addressing the concern about the differentiability of TAGG, the 1-Wasserstein distance used in PDM loss is implemented based on a differentiable optimal transport method by leveraging Gudhi library [15]. Also, the homological feature vector is fed to the MLP layer before incorporating to the node and edge features in the attention module, which ensures the end-to-end training.
>
> ---
>
> **W2) The paper lacks theoretical analysis and does not provide guarantees of topological awareness and preservation.**
>
> **A)** Based on the theoretical foundations of algebraic topology [5], topology-awareness given to the model allows topologically aligned generation of graphs, e.g., the number of connected components and loops. Specifically, the homological features encoded across the entire filtration of a graph hold topological information which cannot be observed in the adjacency matrix of the original graph, providing a rich structural information to the denoising network.
>
> Also, minimizing the distance between the bijection of topological features in the generated and original graphs, as used in the PDM loss, enhances the preservation of topological features. This phenomena is empirically demonstrated in Figure 6 of our paper where the mean landscape generated from TAGG best matches that of the test set.
>
> ---
>
> **W3) Computational overhead needs to be addressed on computing persistent homology.**
>
> **A)** Please refer to the **3. Computation and practical feasibility** part of the global comments.
>
> ---
>
> **W4) What is the fundamental difference of TAGG from previous applications?**
>
> **A)** As the reviewer pointed out, the use of persistent homology in graph analysis has been explored in previous works as per related work section, which focus on optimizing the filter function (e.g., GFL [6]) and utilizing the vectorized homological features simply as additional inputs to the network for downstream tasks (e.g., PersLay [13], GEFL [14]). To the best of our knowledge, our approach is the first to incorporate persistent homology for the graph generation task. Specifically, by utilizing the homological feature $\mu$ and the PDM loss, our method is able to achieve topological preservation throughout the generation process. Importantly, as shown in our results, this topological preservation directly contributes to improved generation performance.
>
> Therefore, while persistent homology has been previously applied in graph analysis, our work represents a meaningful advancement on the graph generation task.

---

> ### Author Response · Authors · 2024-11-21
>
> **Regarding your questions:**
>
> ---
>
> **Q1) What are the main scalability challenges on large graphs, including computational overhead?**
>
> **A)** Please refer to the **3. Computation and practical feasibility** part of the global comments.
>
> ---
>
> **Q2) How sensitive is the performance to the choice of filtration function?**
>
> **A)** We additionally conducted further experiments with alternative filter functions, i.e., the clustering coefficient and betweenness centrality. As shown in Table 5 in the updated Appendix D.3, changing the filter function does affect the optimization process; for instance, using the clustering coefficient as the filter function improves its corresponding MMD score. However, the overall change in general performance is minor, as the new averaged MMD metrics do not diminish the superior performance compared to baseline methods.
>
> ---
>
> **Q3) Can TAGG be trained in an end-to-end manner?**
>
> **A)** Addressing the differentiability aspect: we utilize the pre-computed homological feature $\mu$ as input to the denoising network. Our proposed self-attention module incorporates a simple, differentiable MLP layer, and our proposed PDM loss is computed based on Gudhi library [15], ensuring differentiability. Thus, our proposed method is fully differentiable and enables end-to-end training.
>
> ---
>
> **Q4) Can the learned hidden representations of TAGG be illustrated?**
>
> **A)** From a topological perspective, Figures 5 and 6 provide a clear visualization of the trained homological features, where TAGG shows the most improved discrepancy with the test graphs. Based on these observations, we conducted an additional analysis by visualizing the newly trained features from the topology-aware attention module in a 2-dimensional plane using t-SNE (Figure 11 in the updated Appendix E).
>
> It is important to note that incorporating either the homological feature $\mu$, the PDM loss, or both consistently improved performance across all datasets. In line with the enhanced quantitative results, the visualization reveals differences in the latent space between the cases where neither method was applied and where both were utilized, indicating that the graph features were optimized into a more desirable latent space.
>
> ---
>
> **Q5) Could TAGG be extended to dynamic graphs?**
>
> **A)** Thank you for this insightful question. Currently, our method is designed specifically for static graphs.
>
> Dynamic graphs, by nature, do not evolve as a sequence of nested subgraphs, which is a fundamental requirement for defining a filtration. Consequently, treating dynamic graphs as a type of filtration is not feasible. Furthermore, to the best of our knowledge, no prior work has applied persistent homology to dynamic graphs, likely due to the substantial computational cost involved. Nevertheless, the conceptual similarity between the growth of a smaller simplicial complex into a larger one and the evolution of dynamic graphs presents an intriguing avenue for future research.
>
> We see this as a valuable direction for future research, and exploring methods to optimize or approximate graph filtrations in a dynamic setting could offer promising solutions. Thank you for inspiring this potential extension of our work.

---

> ### Author Response · Authors · 2024-11-25
>
> Dear reviewer 6rDp,
>
> We sincerely appreciate your thorough and constructive reviews. We have carefully addressed your concerns and made revisions accordingly. As the discussion period is nearing its end, we kindly request you to confirm whether our rebuttal has addressed your concerns and allow us the chance to respond to any further questions or feedback. We would be happy to address them and incorporate your comments to improve the quality of the paper. We will be available until the remaining rebuttal period and we will try our best to get the paper accepted.
>
> Sincerely, Authors of 6396

---

> ### Comment · Area_Chair_JEa9 · 2024-11-25
>
> Could please acknowledge and respond to the rebuttal.

---

> ### Comment · Reviewer_6rDp · 2024-11-27
>
> Thank you for the response.
>
> There are still some concerns left.
>
> In general, the design of attention mechanism looks ad-hoc. Based on the description of Eq. 8, it is not obvious to see why one should design the attention of a "topology-aware denoising network" like this. It would be beneficial if the author could provide more theoretical (or intuitive) explanations on why the model is designed in that way.
> - For clarification, does the notation $( , )$ in Eq. 8 in general mean concatenation?
> - What is the meaning of $(E_t, \mu_{G_0})$ in $Z_{E_t}$. In general, $E_t$ are edge embeddings, but $\mu_{G_0}$ is a topological invariant. What is the meaning of putting them together as an input of some NN. What output should one expect from such NN.
> Also, what is the meaning of $Z_{V_t}\odot Z_{E_t}$? What kind of features are represented by such function? Why should the node "attention score" and an edge embedding $Z_{E_t}$ combined in this way?
> - What are all these $\mathcal{N}_*$ networks exactly? Are they all the same type of linear transformations or some kind of MLPs?
> - (minor confusion) In the paper $Z_{V_t}$ is called "attention score". Is it more precise that the attention score is referred to the one after Softmax, according to the traditional attention mechanism?

---

> > ### Author Response · Authors · 2024-11-28
> >
> > We sincerely appreciate your request for clarification and your thoughtful feedback. To thoroughly address your concerns, we have provided more detailed descriptions and revised the PDF accordingly. Please do not hesitate to ask any additional questions or raise further concerns, as we are more than happy to discuss our work and are available at any time.
> >
> > ---
> >
> >
> > * **Explanations on why the "topology-aware denoising network" is designed this way.**
> >
> > **A)** Persistent homology encodes the global structural information of a graph, which is highly relevant to the connectivity and the edge features in a graph. Therefore, the homological feature $\mu_{G_0}$ is incorporated into the edge feature $E_t$ in the attention module, enabling the subsequent neural network to encode this global structural information into the edge embeddings. This topology-aware edge embedding allows the attention module of the denoising network to estimate topology-aware node and edge embeddings, ultimately facilitating the estimation of the topology-aware probability vector.
> >
> > The effectiveness of this approach is empirically proven in Figure 11 of the updated Appendix E, where the projection of the resultant features from the attention module with and without $\mu_{G_0}$ shows a clear distinction. Furthermore, the ablation study on Section 5.2 highlights meaningful performance improvement, underscoring the utility of the topology-aware attention module.
> >
> > ---
> >
> > * **Meaning of the notation (,) in Equation 8.**
> >
> > **A)** We apologize for the confusion caused by the notation in Equation 8. The notation ($\cdot$) denotes the input to a neural network, with multiple inputs separated by a comma. Accordingly, ($E_t$, $\mu_{G_0}$) in $Z_{E_t}$ and ($Z_{V_t}$, $Z_{E_t}$) in $E_t$ both serve as inputs to the neural networks $\mathcal{N}_e$ and $\mathcal{N}_E$, respectively. As described in Figure 2, concatenation and element-wise product are applied to the inputs before they are passed to the neural networks. To avoid further confusion, we have updated the PDF accordingly.
> >
> >
> > ---
> >
> > * **What is the meaning of ($E_t$, $\mu_{G_0}$) in $Z_{E_t}$ and $Z_{V_t} \odot Z_{E_t}$?**
> >
> > **A)** $\mu_{G_0}$ encodes the topological features of a graph, providing global structural information to the neural network. When combined with edge feature $E_t$ as input, it enables the network to generate topology-aware edge embeddings that incorporate this global context, information that cannot be inferred from edge features alone. This enriched input allows the neural network to capture additional structural dependencies within the graph, leading to more informed feature representations.
> > Additionally, the element-wise product in $Z_{V_t} \odot Z_{E_t}$ is derived from the graph transformer [16], which introduces injection of the edge information to improve the already computed intermediate attention score $Z_{V_t}$ via element-wise multiplication.
> >
> > ---
> >
> > * **Details of the $\mathcal{N}_*$ networks.**
> >
> > **A)** We are sorry for omitting the implementation details. We follow the traditional attention mechanism, employing neural networks for the query, key, and value, denoted as $\mathcal{N}_q$, $\mathcal{N}_k$, and $\mathcal{N}_v$, respectively. For the neural networks incorporating homological features, $\mathcal{N}_V$, $\mathcal{N}_E$, and $\mathcal{N}_e$, are implemented as simple 2-layer MLPs with ReLU activation. To clarify this, we have updated Figure 2 accordingly.
> >
> > ---
> >
> > * **Terminology for $Z_{V_t}$.**
> >
> > **A)** Thank you for the insight regarding the terminology used for our attention module. Specifically, $Z_{V_t}$ represents the intermediate attention score, as in the traditional attention mechanism. We have revised this term in the updated paper (highlighted in Line 259).
> >
> > ---
> >
> > [16] Dwivedi, Vijay Prakash, et al. "A generalization of transformer networks to graphs." arXiv preprint arXiv:2012.09699 (2020).

---

> > > ### Comment · Reviewer_6rDp · 2024-12-03
> > >
> > > Thanks for the response. My concerns are mostly resolved.
> > >
> > > I have two more questions.
> > >
> > > - The average $\mu$ is used for generation (sampling). What if the dataset has high variance of the distribution of $\mu$? Will it still work?
> > > - In section 4, will the one-hot edge embedding be very large for dense graphs?

---

> > > > ### Author Response · Authors · 2024-12-03
> > > >
> > > > We appreciate your thorough contribution to the rebuttal discussion. We hope the explanation below resolves your concerns.
> > > >
> > > > ---
> > > >
> > > > * **The average $\mu$ is used for generation (sampling). What if the dataset has high variance of the distribution of $\mu$?**
> > > >
> > > > **A)** It is important to understand that the averaging of $\mu$ was performed to obtain the homological features necessary for the inference process in graph generation (Section 4.4), as averaging is commonly employed to represent a distribution of data. Therefore, for datasets with higher variance, the average operation can be replaced with alternative operations (e.g., median or weighted mean) that are robust to variance within the dataset, which is considered a minor modification of our method.
> > > >
> > > > ---
> > > >
> > > > * **In section 4, will the one-hot edge embedding be very large for dense graphs?**
> > > >
> > > > **A)** As described in line 201-205, we consider the node and edge attributes as one-hot vectors. Therefore, the dimension of the edge one-hot vector $e^{i,j}_t$, i.e., the element of $E_t \in \mathbb{R}^{N\times N\times F_E}$, will remain unchanged with respect to the density of a graph.

---

### Official Review · Reviewer_m97e · 2024-11-06

**Soundness:** 3
**Presentation:** 3
**Contribution:** 3
**Rating:** 6
**Confidence:** 3

**Summary:**

This work introduces a persistence diagram matching loss, which aligns the generated graphs’ topology with that of the target, and a topology-aware attention mechanism within the denoising network. By capturing homological structures, this work aims to better replicate global structural characteristics like connectivity and loops, making it suitable for complex applications.

**Strengths:**

1. The proposed idea is interesting.
2. Demonstrated application in real-world brain networks illustrates TAGG’s adaptability to complex, topologically rich datasets.

**Weaknesses:**

1. A thorough complexity analysis of the graph generation process, including the persistent homology component, is essential. Given that this approach may be computationally intensive, providing a comparative analysis with existing baselines would be highly beneficial. This analysis should include a discussion on both time complexity and resource requirements, considering its practical feasibility for large datasets and real-world applications.
2. The paper’s introduction of the topology-aware diffusion approach is distinct. However, a clearer differentiation from [1] is necessary. The authors should emphasize the conceptual advancements over DiGress in terms of preserving global topological features through persistent homology. It would also strengthen the paper to discuss how this method aligns or contrasts with other homology-based approaches in TDA applications for graphs, which are briefly mentioned in the related work section.
3.  Table 3 provides a quantitative comparison between TAGG and baseline models across multiple datasets using MMD metrics. However, the analysis could benefit from a more detailed interpretation. For instance, while TAGG shows improved performance on various datasets, the significance of the metric differences, especially on small-scale graphs, could be expanded upon. Additionally, a discussion on how TAGG’s improvements in clustering and orbit metrics correlate with enhanced topological fidelity would offer a deeper understanding of the model’s strengths.

[1] Vignac, Clement, et al. "Digress: Discrete denoising diffusion for graph generation." arXiv preprint arXiv:2209.14734 (2022).

**Questions:**

See Weakness.

---

> ### Author Response · Authors · 2024-11-21
>
> **W1) A thorough complexity analysis of the graph generation process is required for its practical feasibility for large datasets and real-world applications.**
>
> **A)** Please refer to the **3. Computation and practical feasibility** part of the global comments .
>
> ---
>
> **W2-1) A clearer differentiation from DiGress[3] is required.**
>
> **A)** Thank you for your advice on improving the legibility of our work. Although the discrete diffusion-based generation method [3] demonstrates powerful generation, its objective is designed to estimate the joint distribution of graph nodes and edges, which does not consider the graph topology at all in the generated graphs, i.e., homological features of a graph. TAGG provides an enhanced denoising network and a topological regularizer via persistent homology, which yields a significant difference in generation performance via the lens of graph topology.
>
> **W2-2) How does TAGG align or contrasts with other homology-based approaches in TDA applications for graphs?**
>
> **A)** To further elaborate the differentiation of TAGG from existing TDA applications for graphs, we provide additional information of the related works, which were not included in the paper due to page limit. Existing homology-based approaches concentrate either on improving the persistent homology representation, e.g., training the filter function using neural networks instead of using predefined heuristic methods [6], or on naive utilization of the vectorized homological features as additional inputs to the downstream classifiers, e.g., feeding homological embedding obtained through MLP layers to the classifier for graph classification [13-14]. Moreover, prior graph generation approaches [1-4] primarily aim to resemble the joint distribution of the node and edge, often overlooking the topological characteristics, such as connected components and loops, which are crucial for graph fidelity.
>
> However, TAGG utilizes homological features in two ways:
> **(1)** a topology-aware attention module, which enables the denoising network to encode topological features, and
> **(2)** a persistence diagram matching loss, serving as a regularizer to promote graph generation with high topological fidelity.
>
> ---
>
> **W3) More detailed interpretation of Table 3, including how TAGG’s improvements in clustering and orbit metrics correlate with enhanced topological fidelity, would be beneficial.**
>
> **A)** In response to your suggestion, due to the page limit of our paper, we added an experimental analysis section in the Appendix D., with revised contents highlighted. As shown in Table 3, the use of the PDM loss generally improves the metrics across various datasets including small and large real-world datasets. These improvements in overall metrics indicate that TAGG can effectively generate the high-fidelity graphs due to the PDM loss, which matches the persistent diagram of the estimated graph with that of the original graph, ensuring topological preservation during the training process of the denoising network.
>
> Additionally, the improved clustering and orbit metrics demonstrate that PDM loss can effectively help the models to preserve the topological structures. Specifically, the improvements in the clustering metric emphasize the TAGG’s ability to preserve the local connectivity patterns, while the improved orbit metric, which reflects the structural patterns of each node's local subgraph, indicates that TAGG effectively preserves the structural roles and relationships within the graph. These results confirm that the PDM loss significantly contributes to generating high-fidelity graphs.

---

> ### Author Response · Authors · 2024-11-25
>
> Dear reviewer m97e,
>
> We sincerely appreciate your thorough and constructive reviews. We have carefully addressed your concerns and made revisions accordingly. As the discussion period is nearing its end, we kindly request you to confirm whether our rebuttal has addressed your concerns and allow us the chance to respond to any further questions or feedback. We would be happy to address them and incorporate your comments to improve the quality of the paper. We will be available until the remaining rebuttal period and we will try our best to get the paper accepted.
>
> Sincerely, Authors of 6396

---

> ### Comment · Area_Chair_JEa9 · 2024-11-25
>
> Could please acknowledge and respond to the rebuttal.

---

### Official Review · Reviewer_aEEb · 2024-11-10

**Soundness:** 3
**Presentation:** 2
**Contribution:** 3
**Rating:** 5
**Confidence:** 4

**Summary:**

This paper introduces a novel graph generation method that utilizes persistent homology within a diffusion model, enhancing structural characteristics. The proposed Persistence Diagram Matching (PDM) loss, in conjunction with the 1-Wasserstein distance, significantly improves the model's topological awareness. The approach extends its application to complex brain network data, demonstrating its effectiveness in real-world scenarios.

**Strengths:**

Pros

1. The proposed method differs from traditional self-attention models by incorporating persistent homology encoding and the use of Q in the forward process, addressing the structural characteristics of graphs.
2. The denoising model integrates the 1-Wasserstein distance to introduce a new loss function (PDM), which is highly versatile and effectively enhances the model's structural awareness.
3. The application range is extended from traditional graphs to brain data with structural features, demonstrating the method's effectiveness on complex real-world datasets.

**Weaknesses:**

Cons

1. As a self-attention model, the paper does not clarify the costs associated with introducing attention mechanisms with mu and training multiple models.
2. The evaluation lacks diverse metrics for comparing graph properties, limiting a comprehensive assessment of model performance.
3. As a generative model, it should generate similar graphs without prior conditions. Here, mu and Q are given to generate the graph, this would naturally make the graph tend to have those similar mu and Q? If so, how to obtain the mu and Q for the graph to be generated? If mu and Q of the test graph themselves are used during the test process, this will make the generated result naturally close to the original graph. If use training ones this would lead to bias?
4. While the loss function emphasizes degree distribution, this leads to the model showing the most significant improvements in degree metrics, while performance on other indicators is less pronounced.

**Questions:**

Please refer to Cons.

---

> ### Author Response · Authors · 2024-11-21
>
> **W1) The paper does not clarify the costs of introducing attention mechanisms with $\mu$ and training multiple models.**
>
> **A)** As mentioned in Section 4.2, the homological feature $\mu$ for graphs in the training dataset are pre-computed, resulting in only a minimal additional cost from training a simple MLP ( $\mathcal{N}_e​$ in Equation 8). Therefore, our proposed module enables the inclusion of supplementary structural information, i.e., $\mu$, with minimal impact on model efficiency.
>
> Note that this inclusion significantly enhances the model’s structural feature awareness, which consistently improves the performance as reported in Table 2. Additionally, there seems to be a misunderstanding: we do not train multiple models. Instead, we train a single model with multiple attention modules, which does not require additional computation compared to other transformer-based methods.
>
> ---
>
> **W2) The evaluation lacks diverse metrics for comparing graph properties.**
>
> **A)** To compare graph generation performance, we used the MMD-based metric, widely adopted in previous works [1-4] and well-validated as a quantitative standard for evaluating the structural characteristics of generated graphs [11]. To further demonstrate the impact of the proposed topology-aware learning framework on the homological similarity of generated graphs, we provided  TDA-based qualitative evaluations of homological features in the main paper (Figures 5 and 6). We believe these experiments address your concerns, as they provide a clear and effective visualization of the homological similarities.
>
> Specifically, ATOL, shown in Figure 5, projects the homological feature vectors on a 2-dimensional plane, which visualizes the proximity of homological features between the test and generated graphs. Also, the mean landscape, shown in Figure 6, encodes the topological summary of a graph, enabling a visual comparison of discrepancies. Compared to the baseline methods, both ATOL and the mean landscape results of TAGG show significant alignment to the test graphs, demonstrating that our proposed topology-aware self-attention module and PDM loss effectively increase the structural awareness of the denoising network.
>
> ---
>
> **W3-1) Does providing $\mu$ and $Q$ naturally bias the generated graph toward similar $\mu$ and $Q$?**
>
> **A)** It seems there might be a slight misunderstanding of our method, and we are sorry for the confusion. The $\mu$ is a pre-computed persistence landscape vector from the training graph, and $Q$ is a noise model applied to the original graph $G_0$​ in discrete space.
>
> Starting with the original graph $G_0$​, we utilize the pre-computed $\mu_{G_0}$ and apply $Q_t$​ to create the noised graph $G_t$​. Given $\mu_{G_0}​​$ and $G_t$​, our proposed denoising network estimates $\hat{G_0}$​, learning to resemble the $G_0​$ through the  cross-entropy loss and PDM loss, i.e., Equation 11. Thus, the model is not trained to match or generate graphs similar to $\mu$ or $Q$.
>
> **W3-2) How are $\mu$ and $Q$ obtained for generation?**
>
> **A)** It is important to understand that the use of the average $\mu$ is essential to ensure that the denoising network incorporates the necessary homological feature $\mu$ even during inference. This approach is a practical necessity rather than a biasing factor, as it allows the model to operate without direct access to the specific $\mu$​​ of any generated sample.
>
> Specifically, we utilize the averaged $\mu$ across all training graphs during inference, as generating new graphs from pure noise samples does not provide $\mu$ for the generated graphs. Moreover, using the average of $\mu$ across the entire training set ensures generality and minimizes dependency on individual graphs within the training set.
>
> ---
>
> **W4) Performance on metrics other than the degree metric is less pronounced, as the loss function emphasizes the degree distribution.**
>
> **A)** Thank you for your insight. We selected the degree function as a filter function, as it is one of the most simple but common and effective ways to define the filtration of a graph [6,12-13]. Notice that this is necessary to obtain the homological feature $\mu$. As the reviewer pointed out, while this degree-based filter function may lead to relatively less improvement for the clustering and orbit metrics compared to degree metric, the average results of TAGG on overall datasets are consistently improved compared to other baselines.
>
> Additionally, we conducted further experiments with alternative filter functions, i.e., the clustering coefficient and betweenness centrality. As shown in Table 5 in the updated Appendix D.3, changing the filter function does affect the optimization process; e.g., using the clustering coefficient as the filter function improves its corresponding MMD score. However, the overall change in general performance is minor, as the new averaged MMD metrics do not diminish the superior performance compared to baseline methods.

---

> > ### Comment · Reviewer_aEEb · 2024-11-25
> >
> > Thanks for your response. So far, I want to retain my score, but I am not against the acceptance if all other reviewers are ok with the paper. Thanks.

---

> ### Author Response · Authors · 2024-11-25
>
> Dear reviewer aEEb,
>
> We sincerely appreciate your thorough and constructive reviews. We have carefully addressed your concerns and made revisions accordingly. As the discussion period is nearing its end, we kindly request you to confirm whether our rebuttal has addressed your concerns and allow us the chance to respond to any further questions or feedback. We would be happy to address them and incorporate your comments to improve the quality of the paper. We will be available until the remaining rebuttal period and we will try our best to get the paper accepted.
>
> Sincerely, Authors of 6396

---

### Official Review · Reviewer_zNwP · 2024-11-11

**Soundness:** 3
**Presentation:** 2
**Contribution:** 2
**Rating:** 5
**Confidence:** 3

**Summary:**

This paper introduces a topology-aware diffusion-based method for realistic graph generation, aiming to preserve structural characteristics similar to the original graphs. The authors propose a novel approach that integrates topological data analysis (TDA), specifically using persistent homology, to guide the generation process. They introduce a new loss function Persistence Diagram Matching (PDM) loss that ensures the generated graphs closely align with the topology of the original graphs, thereby improving fidelity and preserving essential homological properties. Additionally, a topology-aware attention mechanism is developed to enhance the self-attention module in the denoising network.

**Strengths:**

1. This paper addresses the challenge of existing graph generation methods failing to preserve topological information by proposing  a novel topology-aware graph generation method that yields homologically similar graphs with high fidelity.
2. This paper demonstrates the effectiveness of the proposed approach through comprehensive experiments, exhibiting high generation performance across various metrics and aligning better with the distribution of topological features observed in the original graphs.

**Weaknesses:**

1. The paper does not adequately address the relationship between the Preliminaries and the Methodology, making it challenging to follow. Section 3 introduces many topological concepts, which can be frustrating if understanding them is required to proceed with reading. However, Section 4 did not specifically clarify how to apply these topological concepts. This gives me the feeling that there is no need to understand the various topological concepts mentioned in Section 3.
2. Motivation is not convincing enough. The main contribution of this paper is to maintain the topological features of the original graph during the graph generation process. However, it does not analyze the existing methods' capability in capturing topological invariance characteristics.
3. The paper focuses on graph structure information and proposes a topology-aware diffusion-based graph generation method. However, the generalization ability to small graph data: the performance improvement of the model on small graph datasets (e.g., Ego-small) is not obvious, which may indicate that the generalization ability of the model on small graph data needs to be improved. Although the model aims to recover the original graph structure from noisy graphs, how sensitive and robust it is to noise is not detailed in the text. In practice, different noise types and intensities may have an impact on model performance.

**Questions:**

Please refer to weaknesses.

---

> ### Author Response · Authors · 2024-11-21
>
> **W1) The paper does not adequately address the relationship between the Preliminaries(Section 3) and the Methodology(Section 4).**
>
> **A)** Utilizing persistent homology in the graph generation task is motivated by the theoretical insights from algebraic topology, which can extract global structural features of an object in a topological space. However, as persistent homology remains relatively unfamiliar within the machine learning community and to provide theoretical support of our method, it was essential to include the very fundamental concepts in the preliminary section (Section 3).
>
> Also, it is very important to understand the topological concepts to fully understand our method. For instance, L264-266 highlights the advantage of utilizing the "filtration" of a graph over a simple adjacency matrix (or original graph). We were worried that without a clear understanding of filtration, a key concept in persistent homology, readers may find it challenging to understand the theoretical impact of our method. Thus, the preliminary section not only facilitates a clearer understanding of our method, but also elucidates the novel theoretical contributions that distinguish our approach from existing baselines. Below, we pinpoint where the preliminary material is used in our paper.
>
> **Preliminaries in our main method**
> We utilize two major concepts introduced in Section 3, which are the persistence diagram (Eq. 4) and the persistence landscape (Eq. 6), each in the topology-aware attention module (Section 4.2) and the PDM loss (Section 4.3) of TAGG respectively. Both the persistence diagram and the persistence landscape encode homological features of a graph derived from a filtration sequence (Eq. 3 in Section 3), capturing the emergence and persistence of topological features within nested simplicial complexes, i.e., subgraphs.
>
> **Preliminaries in our experiment**
> Such representation methods for persistent homology are further developed in Section 5.3, i.e., ATOL and Mean Landscape, for qualitative assessment of the trained homological features. ATOL and Mean Landscape plot the homological feature vectors obtained from the persistence diagrams (Eq. 4 in Section 3) and the averaged function across the piecewise-linear functions (Eq. 5 in Section 3) on a 2-dimensional plane, respectively.
>
> In summary, our method leverages the persistence diagram and persistence landscape to generate high-fidelity graphs. Therefore, the concepts introduced in the preliminary, such as simplicial complex, filtrations and the birth and death of homological features, are essential components to fully describe TAGG, make the paper self-contained and provide the theoretical basis for our method.

---

> ### Author Response · Authors · 2024-11-21
>
> **W2) Lack of analysis on the existing methods capability in capturing topological invariant characteristics.**
>
> **A)** Existing graph generation methods typically focus on approximating the joint distribution of graph nodes and edges [1-4]. However, from the perspective of persistent homology, these methods rely solely on the final subgraph within the filtration of a graph, i.e., the original graph, overlooking the richer topological information embedded across the entire filtration sequence. Consequently, these existing methods are limited in their ability to comprehensively capture the topological invariant characteristics of graphs compared to TAGG, which utilize the entire filtration of a graph.
>
> Examples illustrating these limitations are presented in Figures 3 and 4. For the brain network (Figure 3), the test dataset has a single connected component due to the inter-hemisphere connectome, a subtle yet critical factor for fidelity. Despite achieving reasonable MMD scores by optimizing the joint distribution of graph nodes and edges (as shown in Table 1), baseline methods fail to generate topologically similar brain networks, e.g., the number of connected components or the sparsity of the brain network. This is also evident in Figures 5 and 6, where significant discrepancies in ATOL and mean landscape can be observed.
>
> Additionally, even the most realistic result, produced by DiGress, fails to replicate the inter-hemisphere connections, resulting in two connected components. A similar phenomenon is observed in the ENZYMES dataset (Figure 4), where the baseline methods fail to resemble the topological properties of the test graphs. The discrepancy of topological features between the reference graphs and the generated graphs of baseline methods can also be observed in Figure 5 and 6.
>
> Our method, on the other hand, captures homological features throughout the filtration, enabling a more topology-aware generation process. Theoretical insights from algebraic topology allow us to capture these global structural features within a graph’s topological space [5]. These features represent an enhanced topological information across every subgraph in the filtration, which cannot be achieved when observing only the final subgraph as in existing methods.
>
> Moreover, previous studies utilizing persistent homology in graph tasks [6,7] have demonstrated the effectiveness of topological features in capturing the richer representation, supporting our motivation to apply persistent homology to graph generation tasks, where capturing structural fidelity is pivotal.
>
> A good example can be seen in Figure 3, where every baseline method either fails to generate brain structure or neglects essential topological properties, e.g., the number of connected components.
>
> ---
>
> **W3-1) The generalization ability to small graph data may need to be improved.**
>
> **A)** Due to the small number of simplices in smaller graphs, the number of topological features is inherently limited, which may reduce the impact of our method compared to larger graph datasets. Therefore, the results of each MMD metric (degree, cluster, orbit) in Table 1 may not be significant.
>
> However, the averaged result of the MMD metrics outperforms every baseline on both community-small and ego-small, demonstrating its meaningful impact on generation performance. Moreover, ablation studies shown in tables 2 and 3 both indicate that our proposed method, i.e., the topology-aware self-attention module and PDM loss, consistently leads to performance gain on the overall dataset.
>
> **W3-2) How sensitive and robust it is to noise is not detailed in the text.**
>
> **A)** Inspired by previous works [3,8-10], we selected the uniform transition model, the most simple and commonly used noise model, to ensure consistency and comparability across datasets. For a fair comparison, we applied the same noise model to all datasets and the hyperparameters for each dataset were chosen by a grid search.
>
> While we acknowledge that different noise types and intensities may impact on model performance, exploring such variations lies beyond the scope of our work, as we primarily focus on the impact of persistent homology on the graph generation.

---

> ### Author Response · Authors · 2024-11-25
>
> Dear reviewer zNwP,
>
> We sincerely appreciate your thorough and constructive reviews. We have carefully addressed your concerns and made revisions accordingly. As the discussion period is nearing its end, we kindly request you to confirm whether our rebuttal has addressed your concerns and allow us the chance to respond to any further questions or feedback. We would be happy to address them and incorporate your comments to improve the quality of the paper. We will be available until the remaining rebuttal period and we will try our best to get the paper accepted.
>
> Sincerely, Authors of 6396

---

> ### Comment · Area_Chair_JEa9 · 2024-11-25
>
> Could please acknowledge and respond to the rebuttal.

---

> ### Comment · Reviewer_zNwP · 2024-11-26
>
> Thank you for your detailed response. Overall, I will keep my score.

---

### Author Response · Authors · 2024-11-21
**Comments to all reviewers**

Dear Reviewers,

We would like to express our sincere gratitude for your thorough and constructive reviews. We appreciate the recognition of our contributions, practical applicability, and the validation provided through comprehensive experiments highlighted by the reviewers as:

- **Novel approach :**
  - This paper addresses the challenge of existing graph generation methods … by proposing a *novel topology-aware graph generation method* that yields homologically similar graphs with high fidelity.” (Reviewer zNwP)
  - The proposed method *differs from traditional self-attention models* by incorporating persistent homology encoding and …  a *new loss function (PDM), which is highly versatile and effectively enhances* the model's structural awareness. (Reviewer aEEb)
  - The technical execution of the work is *solid and thorough*. (Reviewer 6rDp)

- **Application on the complex real-world dataset :**
  - Demonstrated application in real-world brain networks illustrates *TAGG’s adaptability to complex, topologically rich datasets*. (Reviewer m97e)
  - The application range is extended from traditional graphs to brain data with structural features, demonstrating the method's *effectiveness on complex real-world datasets*. (Reviewer aEEb)
  - The application to brain network generation is interesting, as *it addresses a real-world problem where topological features are crucial*. (Reviewer 6rDp)

- **Comprehensive experiments :**
  - This paper demonstrates the effectiveness of the proposed approach through *comprehensive experiments*, exhibiting *high generation performance across various metrics* and aligning better with *the distribution of topological features* observed in the original graphs. (Reviewer zNwP)
  - The empirical *evaluation is comprehensive*, with extensive comparisons against baselines and multiple visualizations. (Reviewer 6rDp)
  - The authors also provide *detailed ablation studies*. (Reviewer 6rDp)

---

We also take the reviewers’ concerns seriously, and we will try to fully address all the concerns. We truly hope that our rebuttal clarifies critical issues and helps the reviewers gain a better understanding of our work, possibly leading to a more favorable consideration for the acceptance of this paper. We welcome further discussion during the rebuttal period, so please feel free to leave any comments. We address some common concerns below:


**1. Contribution**

**A)** In our humble opinion, our contributions with respect to the core idea and concept are non-trivial although technical modification may be incremental. The main contribution of our work lies on graph generation with topological fidelity, which is crucial in real-world applications e.g., brain network modeling. To the best of our knowledge, this is the first work to examine graph generation tasks through the lens of persistent homology, validated by comprehensive experiments conducted both qualitatively and quantitatively (which the reviewer pointed as a strength).

Specifically, unlike the existing approaches, our method uniquely captures the homological features across the entire filtration of a graph, rather than focusing solely on the final subgraph, i.e., the original graph. Moreover, our approach incorporates these homological features to ensure topological preservation during the generation process.

As shown in Table. 1 and Figure. 5 and 6, our model generates high-fidelity graphs, with quantitative results (Table. 1) and qualitative results (Figure. 5 and 6). While some baselines perform competitively on MMD metrics, TAGG empirically demonstrates a clear advantage on topology-based metrics, i.e., ATOL and Mean Landscape, highlighting our ability to perverse the topological characteristics.

**2. Experimental analysis**

**A)** We have included additional experimental analyses in Appendix D. Specifically, Appendix D.1, D.2, and D.3 provide a detailed discussion of the ablation studies on the homological feature $\mu$ and PDM loss, the impact of different filter functions, and the generalizability of our method to small and large graph datasets, respectively.

---

> ### Author Response · Authors · 2024-11-21
>
> **3. Computation and practical feasibility**
>
> **A)** We agree with the reviewers’ concern regarding the need for a thorough computation analysis, as persistent homology is particularly known to be expensive. In response, we have detailed the computation analysis below including the graph generation process and persistent homology component. To address the reviewer’s concern, we report the measured sec/epoch and memory usage (in MB) for TAGG and baseline models on four datasets of varying scale. Our experiments are conducted on a single GeForce RTX 3090 with 24GB of GPU memory, with batch size 4.
> |sec / epoch|Ego-small|Comm-small|ENZYMES|ADNI|
> |--|-|-|-|-|
> |DiGress|1.6 (0.03, -)|0.9 (0.02, -)|5.1 (0.09, -)|26.3 (0.17, -)|
> |TAGG|3.0 (0.03, 1.41)|2.4 (0.02, 1.39)|65.0 (0.12, 58.85)|77.1 (0.18, 49.27)|
>
> The table above reports the averaged sec / epoch of 10 epochs. (CE time, PDM time) refers to the time required for calculating the cross-entropy loss and the PDM loss, respectively.
>
> The primary sources of computational burden in TAGG can be decomposed into three aspects: 1) the computation of the homological feature $\mu$, 2) persistent homology, and 3) the Wasserstein distance in the PDM loss. While $\mu$ can be pre-computed, requiring minimum additional computation during training, the computational cost of persistent homology and the Wasserstein distance is comparatively high, accounting for most of the time differences shown in the table above. This limitation is primarily due to the CPU-based implementation of both methods, a significant challenge faced by researchers in the TDA community, as none of the existing Python libraries currently support GPU computation. Despite the well-known high computational complexity of persistent homology, we believe this is still one of the highly attractive research topics to investigate in the graph ML community.
>
> In the case for the graph dataset with >100 nodes, i.e.,  ENZYMES (up to 125 nodes) and real-world ADNI (160 nodes), the number of simplices increase leading to the additional costs for computing the persistent homology, e.g., the graph filtration. Despite this, the training times on ENZYMES and ADNI dataset are 64.8 and 78.1 seconds per epoch, respectively, and fully training the TAGG (1000 epochs) can be done within a day (< 22h) on both datasets.
> | sec| Ego-small | Comm-small | ENZYMES | ADNI  |
> |--|--|--|--|--|
> |DiGress|13.92|14.27|14.48|63.18|
> |TAGG|14.99|14.44|15.04|64.19|
>
> To verify the practicality of our model, we also report the average inference time (sec) for generating 4 graph samples over 5 independent runs across all datasets in the table above, using a single GPU (GeForce RTX 3090 with 24GB). As shown above, the inference time of TAGG is comparable to that of the baseline model, i.e., Digress, across all datasets. This demonstrates that TAGG achieves superior performance and fidelity in graph generation, as reported in Section. 5, without incurring any significant additional computational cost.
>
> In addition, as shown in the table below, the memory requirements for TAGG are not substantial. Considering that these results were achieved using a single GPU, we think that the computational demand is practically feasible for real-world applications especially when multiple GPUs are employed.
> |MB|Ego-small|Comm-small|ENZYMES|ADNI|
> |--|--|--|--|--|
> | DiGress|1532|1570|5988|5680|
> |TAGG|1542|1580|6352|6186|
>
> In summary, we believe that the added complexity of TAGG, as reported in the tables above, is justified by its improved performance on topologically rich datasets, making it a practical and valuable tool for applications requiring high topological fidelity.

---

> ### Author Response · Authors · 2024-11-21
>
> **[References for rebuttal comments]**
>
> [1] Jo, Jaehyeong, et al. "Score-based generative modeling of graphs via the system of stochastic differential equations." ICML, 2022.
> [2] Niu, Chenhao, et al. "Permutation invariant graph generation via score-based generative modeling." AISTATS, 2020.
> [3] Vignac, Clément, et al. "DiGress: Discrete Denoising diffusion for graph generation." ICLR, 2023.
> [4] Bergmeister, Andreas, et al. "Efficient and Scalable Graph Generation through Iterative Local Expansion." ICLR, 2024
> [5] Edelsbrunner, Herbert, et al. “Computational topology: an introduction.” American Mathematical Society, 2022.
> [6] Hofer, Christoph, et al. "Graph filtration learning." ICML, 2020.
> [7] Horn, Max, et al. "Topological graph neural networks." ICLR, 2022.
> [8] Hoogeboom, Emiel, et al. "Argmax flows and multinomial diffusion: Learning categorical distributions." NeurIPS, 2021.
> [9] Austin, Jacob, et al. "Structured denoising diffusion models in discrete state-spaces." NeurIPS, 2021.
> [10] Yang, Dongchao, et al. "Diffsound: Discrete diffusion model for text-to-sound generation." TASLP, 2023.
> [11] Thompson, Rylee, et al. "On Evaluation Metrics for Graph Generative Models." ICLR, 2022.
> [12] Hofer, Christoph, et al. "Deep learning with topological signatures." NeurIPS, 2017.
> [13] Carrière, Mathieu, et al. "Perslay: A neural network layer for persistence diagrams and new graph topological signatures." AISTATS, 2020.
> [14] Zhang, Simon, et al. "GEFL: extended filtration learning for graph classification." Learning on Graphs Conference. PMLR, 2022.
> [15] Maria, Clément, et al. "The gudhi library: Simplicial complexes and persistent homology." Mathematical Software–ICMS, 2014

---

### Note · Authors · 2025-01-31

I have read and agree with the venue's withdrawal policy on behalf of myself and my co-authors.